# A shooting formulation of deep learning

**François-Xavier Vialard**
LIGM, Univ. Gustave Eiffel, CNRS
francois-xavier.vialard@u-pem.fr

**Roland Kwitt**
Department of Computer Science
University of Salzburg
Roland.Kwitt@sbg.ac.at

**Susan Wei**
School of Mathematics and Statistics
University of Melbourne
susan.wei@unimelb.edu.au

**Marc Niethammer**
Department of Computer Science
University of North Carolina at Chapel Hill
mn@cs.unc.edu

## Abstract

A residual network may be regarded as a discretization of an ordinary differential equation (ODE) which, in the limit of time discretization, defines a continuous-depth network. Although important steps have been taken to realize the advantages of such continuous formulations, most current techniques assume *identical* layers. Indeed, existing works throw into relief the myriad difficulties of learning an infinite-dimensional parameter in a continuous-depth neural network. To this end, we introduce a shooting formulation which shifts the perspective from parameterizing a network layer-by-layer to parameterizing over *optimal* networks described only by a set of *initial conditions*. For scalability, we propose a novel particle-ensemble parameterization which fully specifies the optimal weight trajectory of the continuous-depth neural network. Our experiments show that our particle-ensemble shooting formulation can achieve competitive performance. Finally, though the current work is inspired by continuous-depth neural networks, the particle-ensemble shooting formulation also applies to discrete-time networks and may lead to a new fertile area of research in deep learning parameterization.

## 1 Introduction

Deep neural networks (DNNs) are closely related to optimal control (OC) where the sought-for control variable corresponds to the parameters of the DNN [24, 23, 19]. To be able to talk about an *optimal* control requires the definition of a control cost, i.e., a norm on the control variable. We explore the ramifications of such a control cost in the context of DNN parameterization. For simplicity, we focus on continuous formulations in the spirit of neural ODEs [13]. However, both discrete and continuous OC formulations exist [12, 4, 37]; our approach could be developed for both.

Initial work on continuous DNN formulations was motivated by the realization that a `ResNet` [20, 21] resembles Euler forward time-integration [19, 23]. Specifically, the forward pass of some input vector $\tilde{\mathbf{x}} \in \mathbb{R}^d$ through a network with $L$ layers, specified as $\mathbf{x}(0) = \tilde{\mathbf{x}}$ and $\mathbf{x}(j + 1) = \mathbf{x}(j) + f(\mathbf{x}(j), \theta(j))$, $j = 0, 1, \dots, L$, closely relates to an explicit Euler [36] discretization of the ODE

$$\dot{\mathbf{x}}(t) = f(t, \mathbf{x}(t), \theta(t)), \quad \mathbf{x}(0) = \tilde{\mathbf{x}}, \quad 0 \leq t \leq T \ . \quad (1.1)$$

In the continuous DNN formulation, we seek an optimal $\theta$ such that the terminal prediction given by $\mathbf{x}(T)$, i.e., the solution to Eq. (1.1) at time $T$, minimizes $\ell(\mathbf{x}(T))$ for a task-specific loss function $\ell$.

Although Eq. (1.1) with time-varying parameter $\theta(t)$ can be considered as a neural network with an infinite number of layers, current implementations of ODE-inspired networks largely assume the

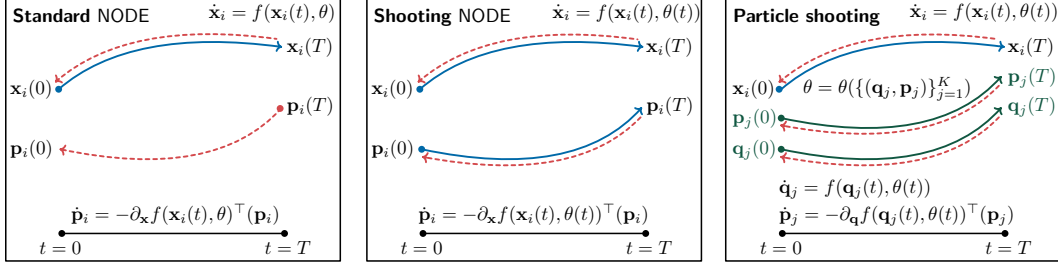

**Figure 1:** Optimization in the neural ODE (NODE) framework [13] (left) amounts to a forward pass with the gradient computed via backpropagation (--). Optimization under the shooting principle (middle) turns the forward-backward system into a forward *second-order* system, where we essentially run the backpropagation equation forward. We use a Hamiltonian particle ensemble (right) consisting of $K$ (position, momentum) pairs $(\mathbf{q}_j, \mathbf{p}_j)$ to make shooting efficient. Note that we write $\theta = \theta(\{(\mathbf{q}_j, \mathbf{p}_j)\}_{j=1}^{K})$ since $\theta$ satisfies a compatibility equation which involves all $K$ particles. In shooting $\theta$ is time-dependent, in standard NODE $\theta(t) = \theta \, \forall t$.

parameters $\theta$ are fixed in time, i.e., $\forall t : \theta(t) = \theta$ [13, 14], or follow some prescribed dynamics [44]. Instead, we explore time-varying $\theta(t)$ by employing regularization (i.e., a control cost) to render the estimation well-posed and to assure regularity of the resulting flow. Specifically, (for a single data point) we propose minimizing over $\theta$ the regularized loss

$$\mathcal{E}(\theta) = \int_0^T R(\theta(t)) \, \mathrm{d}t + \gamma \, \ell(\mathbf{x}(T)), \quad \gamma \in \mathbb{R}^+, \quad \text{subject to Eq. (1.1)} , \qquad (1.2)$$

where $R$ is a real-valued complexity measure of $\theta$ corresponding to the control cost. We will mostly work with the Frobenius norm but $R(\theta(t))$ can be more general (see Appendix B).

Instead of *directly* optimizing over the set of time-dependent $\theta(t)$ as in standard `ResNets`, *we restrict the optimization set to those $\theta$ which are critical points of $\mathcal{E}(\theta)$, thereby dramatically reducing the number of parameters.* In doing so, one can describe the optimization task as an *initial value problem*. Namely, we show that we can rewrite the loss in Eq. (1.2) solely in terms of the input $\mathbf{x}(0)$ and a corresponding finite-dimensional momentum variable, $\mathbf{p}(0)$. Such an approach, just like optimizing the initial speed of a mass particle to reach a given point, is called a *shooting method* in numerical analysis [30] and control [10], giving its name to our new formulation.

The first two panels of Fig. 1 illustrate the difference between the optimization of a neural ODE (NODE) via [13] and our shooting formulation. Since in practice, we have multiple inputs $\tilde{\mathbf{x}}_i, i = 1, \dots, n$, there is an initial momentum vector $\mathbf{p}_i$ corresponding to each of them. If the shooting formulation is to scale up to a large sample size $n$, we must take care that the parameterization does not grow linearly with $n$. To this end, we propose what we call the *Hamiltonian particle-ensemble parameterization*. It is a finite set of particles, where each particle is a (position, momentum) pair. The initial conditions of these particle pairs $\{(\mathbf{q}_j, \mathbf{p}_j)\}_{j=1}^{K}$ (where $K \ll n$) completely determine $\theta(t)$. This is illustrated in the rightmost panel of Fig. 1. Once the optimized set of particles has been computed, the computational efficiency of the forward model, similarly to NODE [13], is retained for vector fields $f$ that are linear in their parameters $\theta(t)$.

Our **contributions** are as follows: 1) We introduce a shooting formulation for DNNs, amounting to an initial-value formulation for neural network parameterization. This allows for optimization over the original network parameter space via optimizing over the initial conditions of critical networks only; 2) We propose an efficient implementation of the shooting approach based on a novel particle-ensemble parameterization in which a set of initial particles (the (position, momentum) pairs) describe the space of putative optimal network parameters; 3) We propose the `UpDown` model which gives rise to explicit shooting equations; 4) We prove universality for the flows of the `UpDown` vector field and demonstrate in experiments its good performance on several prediction tasks.

## 2   Related work

We draw inspiration from two separate branches of research: 1) continuous formulations of neural networks [13] and 2) shooting approaches for deformable image registration [38, 27, 29].

**Continuous-depth neural networks.**  Continuous equivalents of `ResNets` [20, 21] have been developed in [32, 19], but naïve implementations are memory-demanding since backpropagation

requires differentiating through the numerical integrator. Two approaches can address this unfavorable memory footprint. NODE [13] does not store intermediate values in the forward pass, but recomputes them by integrating the forward model backward. This is easily possible only if the forward model is numerically invertible and the formulation is time-continuous [17][1]. Instead, checkpointing [17] is a general approach to reduce memory requirements by selectively recomputing parts of the forward solution [18]. Our work can easily be combined with these numerical approaches.

**Solving implicit equations.** A recent line of works, including deep equilibrium models [6] and implicit residual networks [31], has shown that it may not always be necessary to freely parameterize all the layers in the network. Specifically, in [6] and [31], the parameters of each layer are defined via an implicit equation motivated by *weight tying* thus improving expressiveness and reducing the number of parameters while decreasing the memory footprint via implicit differentiation. Instead, our work increases expressiveness and reduces the number of parameters via particle-based shooting.

**Invertibility and expressiveness.** Based on similarity with continuous time integration, constraining the norm of a layer in a `ResNet` will result in an invertible network such as in [8, 22]. Invertibility is also explored in [42], where it is enforced (as in our setting) via a penalty of the norm. These works show that standard learning tasks can be performed on top of a one-to-one transformation. Recent theoretical developments [43] show that indeed capping a NODE or i-ResNet [8] with a single linear layer gives universal approximation for non-invertible continuous functions. Further, expressiveness can be increased by moving to more complex models, e.g., by introducing additional dimensions as explored in augmented NODE [14]. In [44] (`AnodeV2`), Zhang et al. treat time-dependent $\theta(t)$. Weights are evolved jointly with the state of the continuous DNN. While this weight evolution could, in principle, also be captured by a learned weight network, the authors argue that this would result in a large increase in parameters and therefore opt for explicitly parameterizing these evolutions (e.g., via a reaction diffusion equation). In contrast, our method does not rely on learning a separate weight-network or on explicitly specifying a weight evolution. Instead, our evolving weights are a direct consequence of the shooting equations which, in turn, are a direct consequence of penalizing network parameters (the control cost) over time; a large increase in parameters does not occur.

**Hamiltonian approaches.** Toth et al. [35] proposed Hamiltonian generative networks to learn the Hamiltonian governing the evolution of a physical system. Specifically, they learn Hamiltonian vector fields in the latent space of an image encoder-decoder architecture. Sæmundsson et al. [33] also learn the underlying dynamics of a system from time-dependent data, starting from a discrete Lagrangian combined with a variational integrator. This motivates particular network structures; e.g., Newtonian networks where the potential energy is learned via a neural network. Although sharing common tools, our work completely differs from this line of research in the sense that we exploit Hamiltonian mechanics to parameterize *general* continuous neural networks. In principle, our work applies to most network architectures and is not specific to physical data.

Finally, we mention that shooting approaches have been applied successfully in other areas such as diffeomorphic image matching [27, 38, 29]. However, the decisive difference here is in the dimensionality of the underlying space: in diffeomorphic image registration, the data are points in a 3D volume i.e., $d = 3$; for DNNs applications, data points usually lie in a much higher-dimensional space, i.e., $d$ is very large.

## 3 Shooting formulation of ODE-inspired neural networks

We consider, for simplicity, a supervised learning task where the input and target spaces are $\mathcal{X} \subset \mathbb{R}^d$ and $\mathcal{Y}$, resp., and sampled data are denoted by $\{(\tilde{\mathbf{x}}_i, \tilde{\mathbf{y}}_i)\}_{i=1}^n \subset \mathcal{X} \times \mathcal{Y}$. The goal is to learn the weight $\theta(t)$ in the following flow equation

$$\dot{\mathbf{x}}_i(t) = f(\mathbf{x}_i(t), \theta(t)), \quad \mathbf{x}_i(0) = \tilde{\mathbf{x}}_i, \quad 0 \le t \le T, \quad i = 1, \dots, n \tag{3.1}$$

such that it minimizes the loss $\sum_{i=1}^n \ell(\mathbf{x}_i(T), \tilde{\mathbf{y}}_i)$ for some loss function $\ell$. In existing works, the weight is chosen independent of time, i.e., $\theta(t) = \theta$ [13], or specific evolution equations are postulated for it [25, 44]. Such strategies show the difficulty of addressing infinite dimensional parameterizations of time-dependent $\theta$ and the need for regularization for well-posedness [16, 25, 19].

Instead of parameterizing $\theta(t)$ directly, we aim at penalizing $\theta(t)$ according to the regularity of $f(\cdot, \theta(t))$ to arrive at a well-posed problem. Specifically, we consider a regularization term $R(\theta(t))$ (discussed in §3.1) and propose to minimize over $\theta$

$$\mathcal{E}_n(\theta) = \int_0^T R(\theta(t))\, \mathrm{d}t + \gamma \sum_{i=1}^n \ell(\mathbf{x}_i(T), \tilde{\mathbf{y}}_i), \quad \gamma \in \mathbb{R}^+, \quad \text{subject to Eq. (3.1) .} \tag{3.2}$$

Note that upon discretizing the time $t$ (i.e., having a number of parameters proportional to the number of timesteps) this is similar to a `ResNet` with weight decay. For a `ResNet` or a NODE, optimization is based on computing the parameter gradient via a forward pass followed by backpropagation (see left panel of Fig. 1).

**Optimality equations.** The optimality conditions for Eq. (3.2) in continuous time are:

$$\begin{cases} \dot{\mathbf{x}}_i(t) - f(\mathbf{x}_i(t), \theta(t)) = 0, \ \mathbf{x}_i(0) = \tilde{\mathbf{x}}_i, & \text{Data evolution} \\ \dot{\mathbf{p}}_i(t) + \partial_{\mathbf{x}} f(\mathbf{x}_i(t), \theta(t))^\top (\mathbf{p}_i) = 0, \ \mathbf{p}_i(T) = -\gamma \nabla \ell(\mathbf{x}_i(T), \tilde{\mathbf{y}}_i), & \text{Adjoint evolution} \\ \partial_\theta R(\theta(t)) - \sum_{i=1}^n \partial_\theta f(\mathbf{x}_i(t), \theta(t))^\top (\mathbf{p}_i(t)) = 0\,. & \text{Compatibility} \end{cases} \tag{3.3}$$

The first equation describes evolution of the input data and the second equation is the adjoint equation solved backward in time in order to compute the gradient with respect to the parameters. At convergence, the third equation is also satisfied. This last equation encodes the optimality of the layer at timestep $t$, as it is the case for an *argmin layer* or *weight tying* [2]. Its left hand side corresponds to the gradient with respect to the parameter $\theta$, but as we shall see it will allow us to compute $\theta$ directly via our (position, momentum) pairs in our particle shooting formulation. The shooting approach simply replaces the optimization set by the set of critical points of Eq. (3.2) expressed in these optimality conditions. That is, we only optimize over solutions fulfilling Eq. (3.3).

**Shooting principle.** The shooting method is standard in optimal control [10] and can be formulated as follows: since, at optimality, the system in Eq. (3.3) is satisfied, one can *turn this system into a forward model* defined only by its initial conditions $\{(\mathbf{x}_i(0), \mathbf{p}_i(0))\}_{i=1}^n$ which specify the *entire trajectory* of optimal parameters. We evolve *both* the data and adjoint evolution equations forward in time and compute at each time, $t$, $\theta(t)$ from the compatibility Eq. (3.3) via the current values of $\{(\mathbf{x}_i(t), \mathbf{p}_i(t))\}_{i=1}^n$. We refer to the forward model defined by Eq. (3.3) as the *shooting equations*. Unfortunately, this initial-condition parameterization still requires *all* initial conditions $\mathbf{x}_i(0)$ and their corresponding momenta $\mathbf{p}_i(0)$ for $i = 1, \ldots, n$. Since this does not scale to very large datasets, we propose an approximation using a collection of particles, as described next.

**Hamiltonian particle ensemble.** In the limit and ideal case where the data distribution is known, the optimality equations can be approximated using a collection of particles which follow the Hamiltonian system (see Appendix A). We thus consider a collection of particles $\{(\mathbf{q}_j, \mathbf{p}_j)\}_{j=1}^K \in \mathbb{R}^d \times \mathbb{R}^d$ that drive the evolution of the entire population $\{\mathbf{x}_i\}_{i=1}^n \subset \mathbb{R}^d$ through the following forward model

$$\begin{cases} \dot{\mathbf{x}}_i(t) - f(\mathbf{x}_i(t), \theta(t)) = 0, \ \mathbf{x}_i(0) = \tilde{\mathbf{x}}_i & \text{Data evolution} \\ \dot{\mathbf{q}}_j(t) - f(\mathbf{q}_j(t), \theta(t)) = 0, \\ \dot{\mathbf{p}}_j(t) + \partial_{\mathbf{q}} f(\mathbf{q}_j(t), \theta(t))^\top (\mathbf{p}_j(t)) = 0, \\ \partial_\theta R(\theta(t)) - \sum_{j=1}^K \partial_\theta f(\mathbf{q}_j(t), \theta(t))^\top (\mathbf{p}_j(t)) = 0, \end{cases} \left.\begin{matrix}\\\\\\\\\\\end{matrix}\right\} \text{Hamiltonian equations} \tag{3.4}$$

with initial conditions $\{(\mathbf{q}_j(0), \mathbf{p}_j(0))\}_{j=1}^K$, where the gradient with respect to this new parameterization is computed via backpropagation, and typically $K \ll n$. This set of (position, momentum) pairs is termed the *Hamiltonian particle ensemble*. As the number of particles is reduced, so are the number of free parameters, see Appendix C. Indeed, varying the Hamiltonian particle ensemble allows for controlling the tradeoff between reconstruction and network complexity. Note that the main difference to the shooting formulation of Eq. (3.3) is that the parameterization, $\theta(t)$, is now retrieved from the shooting equations as specified by the particle collection. The original data samples, $\tilde{\mathbf{x}}_i$, are simply propagated via these parameters.

## 3.1 Choices of regularization, parameterization and conserved quantities

The main computational bottleneck in the forward model of Eq. (3.4) is the implicit parameterization of $\theta$ by the last equation. Making it explicit is key to render shooting computationally tractable.

**Linear in parameter[2] - quadratic penalty.** In the simplest case, the space of functions $f$ is a linear space parameterized by $\theta(t)$. In this case, a quadratic penalty amounts to a kinetic penalty. Specifically, as a motivating example, consider the forward model

$$f(\mathbf{x}(t), \theta(t)) = A(t)\sigma(\mathbf{x}(t)) + b(t), \tag{3.5}$$

where $\sigma$ is a component-wise activation function, $A \in L^2([0,1], \mathbb{R}^{d^2})$, $b \in L^2([0,1], \mathbb{R}^d)$ and $\theta(t) = [A(t), b(t)]$. With the quadratic regularizer $R(\theta(t)) = \frac{1}{2}\operatorname{Tr}\left(A(t)^\top M_A A(t)\right) + \frac{1}{2}b(t)^\top M_b b(t)$, where $M_A$, $M_b$ are positive definite matrices, the particle shooting equations are

$$\begin{cases} \dot{\mathbf{q}}_j(t) &= A(t)\sigma(\mathbf{q}_j(t)) + b(t), \\ \dot{\mathbf{p}}_j(t) &= -\operatorname{d}\sigma(\mathbf{q}_j(t))^\top A(t)^\top \mathbf{p}_j(t), \end{cases} \quad \begin{cases} A(t) &= M_A^{-1}(-\sum_{j=1}^K \mathbf{p}_j(t)\sigma(\mathbf{q}_j(t))^\top) \\ b(t) &= M_b^{-1}(-\sum_{j=1}^K \mathbf{p}_j(t)), \end{cases} \tag{3.6}$$

with given initial conditions $(\mathbf{p}_j(0), \mathbf{q}_j(0))$. We emphasize that $\theta(t)$ is *explicitly defined* by $\{(\mathbf{p}_j(t), \mathbf{q}_j(t))\}_{j=1}^K$ and the computational cost is reduced to matrix multiplications.

As is well-known [3], the Hamiltonian flow preserves the Hamiltonian function. In the "linear in parameter - quadratic penalty" case, this preserved quantity, denoted

$$H(\mathbf{p}(t), \mathbf{q}(t)) = R(\theta(t)),$$

corresponds to a (kinetic) energy of the system of particles. As a first consequence, the objective functional can be rewritten as

$$H(\mathbf{p}(0), \mathbf{q}(0))) + \gamma \sum_{i=1}^n \ell(\mathbf{x}_i(T), \tilde{\mathbf{y}}_i).$$

This clearly allows for direct optimization on $(\mathbf{p}(0), \mathbf{q}(0))$, i.e., shooting. As a second consequence, since the vector field has constant norm (its squared norm is the Hamiltonian), it gives a quantitative bound on the regularity of the flow map at time $t = T$ explicit in terms of $H(\mathbf{p}(0), \mathbf{q}(0))$. In addition (Appendix A), the Rademacher complexity of the generated flows with bounded $H(\mathbf{p}(0), \mathbf{q}(0)))$ can also be controlled.

**Nonlinear in parameter and non-quadratic penalty.** A standard `ResNet` structure uses vector fields of the type (in convolutional form or not)

$$f(\mathbf{x}(t), \theta(t)) = \theta_1(t)\sigma(\theta_2(t)\mathbf{x}(t) + b_2(t)) + b_1(t) \ , \tag{3.7}$$

where $\theta_1(t) \in L(\mathbb{R}^{d'}, \mathbb{R}^d)$ and $\theta_2(t) \in L(\mathbb{R}^d, \mathbb{R}^{d'})$. We will refer to Eq. (3.7) as the **single-hidden-layer** vector field. This model can also be handled in our shooting approach since the shooting equations in Eq. (3.3) are completely specified by the Hamiltonian

$$H(\mathbf{p}, \mathbf{q}, \theta) = R(\theta) - \mathbf{p}^\top f(\mathbf{q}, \theta).$$

Automatic differentiation can be used (see Appendix D) to implement the forward model

$$\dot{\mathbf{q}}(t) = \frac{\partial H}{\partial \mathbf{p}}(\mathbf{p}(t), \mathbf{q}(t), \theta(t)), \ \dot{\mathbf{p}}(t) = -\frac{\partial H}{\partial \mathbf{q}}(\mathbf{p}(t), \mathbf{q}(t), \theta(t)), \ \theta(t) \in \arg\min H(\mathbf{p}(t), \mathbf{q}(t), \theta(t)). \tag{3.8}$$

Note that a necessary condition for solving the third equation above is in fact the compatibility equation in Eq. (3.4). Important bottlenecks appear since the third equation is nonlinear and potentially associated with a non-convex optimization problem. This could be addressed by unrolling the optimization corresponding to the last equation, resulting in increased computational cost. In addition, in this nonlinear case, the Hamiltonian function is no longer (in general) equal to $R(\theta(t))$ even in the quadratic regularization setting. Therefore, results on the smoothness or Rademacher complexity would no longer be guaranteed as for the linear - quadratic penalty case. Last, quadratic regularization has no known theoretical results for the Rademacher complexity of functions generated by Eq. (3.7) with bounded norm. Norms for which the Rademacher complexity of this class of functions is known [15] to be bounded are called Barron norms, which are non-smooth and non-convex, and which would add to the difficulty. To circumvent these issues while retaining expressiveness and theoretical guarantees in the linear parameterization setting, we next introduce the `UpDown` model.

## 3.2   The UpDown model

The key idea is to transform the vector field of Eq. (3.7) into a model which is linear in parameters on which the quadratic regularization can be applied. To this end, we introduce the additional state

$$\mathbf{v}(t) = \theta_2(t)\mathbf{x}(t) + b_2(t)$$

which we differentiate with respect to time to obtain

$$\dot{\mathbf{v}}(t) = \dot{\theta}_2(t)\mathbf{x}(t) + \dot{b}_2(t) + \theta_2(t)\dot{\mathbf{x}}(t) \ .$$

Replacing $\dot{\mathbf{x}}(t)$ by its formula, we get

$$\dot{\mathbf{v}}(t) = \dot{\theta}_2(t)\mathbf{x}(t) + \dot{b}_2(t) + \theta_2(t)(\theta_1(t)\sigma(\mathbf{v}(t)) + b_1(t)) \ .$$

Now overloading on notation slightly, we use the additional state variable $\mathbf{v}(t)$ to propose the following ODE system, denoted the UpDown model:

$$\dot{\mathbf{x}}(t) = \theta_1(t)\sigma(\mathbf{v}(t)) + b_1(t), \quad \dot{\mathbf{v}}(t) = \theta_2(t)\mathbf{x}(t) + b_2(t) + \theta_3(t)\sigma(\mathbf{v}(t)), \qquad (3.9)$$

with $\mathbf{x}(t) \in \mathbb{R}^d$, $\mathbf{v}(t) \in \mathbb{R}^{\alpha d}$ and introducing the (integer-valued) *inflation factor* $\alpha \geq 1$. For the data evolution, $\mathbf{x}_i(0)$ are given by the data $\{\tilde{\mathbf{x}}_i\}$. We parameterize the $\mathbf{v}_i(0)$ using an affine map $g_\Theta$, i.e.,

$$\mathbf{v}_i(0) = g_\Theta(\mathbf{x}_i(0)) = \Theta_{12}(\mathbf{x}_i(0)) + b_{12},$$

where $\Theta_{12} \in L(\mathbb{R}^d, \mathbb{R}^{\alpha d})$ and $b_{12} \in L(\mathbb{R}^{\alpha d})$. In Appendix E, we prove the following theorem:

**Theorem 1.** *Given a time-dependent vector field defined on a compact domain $C$ of $\mathbb{R}^d$, which is time continuous and Lipschitz, we denote by $\varphi(T, \mathbf{x}(0))$ its flow at time $T$ from starting value $\mathbf{x}(0)$. Then, there exists a parameterization of the* UpDown *model for which its solution is $\varepsilon$-close to the flow, $\sup_{\mathbf{x}(0) \in C} \|\varphi(T, \mathbf{x}(0)) - \mathbf{x}(T)\| \leq \varepsilon$.*

Notably, in the proof, the dimension of the hidden state $\mathbf{v}$ is used twice: *first*, for having a sufficient number of neurons in Eq. (3.7) to approximate a stationary vector field (standard universality property of multilayer perceptron) and, *second*, for approximating time-dependent vector fields. Therefore, at the cost of introducing a possibly large number of dimensions, the UpDown model is universal in the class of time-dependent NODEs. As shown in Appendix E, this universality result transfers to our shooting formulation. Due to its additional dimensions, it is also likely to be universal in the space of functions (i.e., not necessarily injective). We focus on the UpDown model in our experiments. Note also that while we derived our theory for vector-valued evolutions for simplicity, similar linear in parameter evolution equations can for example be derived for convolutional neural networks.

## 4   Experiments

Our goal is to demonstrate that it is possible to learn DNNs by optimizing only over the initial conditions of *critical* networks. This is made possible via shooting and efficient via our particle parameterization. A key difference to prior work is that our approach allows to capture time-dependent (i.e., layer-dependent in the discrete setting) parameters *without* discretizing these parameters at every time-point. Comparisons to other NODE like methods are not straightforward due to hyperparameters and different implementations. For consistency, we therefore provide four different formulations (based on the UpDown model of §3.2).

- The **static direct** model forgoes the Hamiltonian particle ensemble, and instead directly optimizes over *time-constant* parameters: $\theta(t) = \theta$ for all $t$. Everything else, including the UpDown model, stays unchanged. This model is most closely related to NODE [13] and augmented NODE [14].

- We call our proposed shooting model **dynamic with particles**. It is parameterized via a set of initial conditions of (position, momentum) pairs, which evolve over time and fully specify $\theta(t)$.

- The **static with particles** model is similar to the *static direct* model. However, instead of directly optimizing over a *time-constant* $\theta$, it uses a set of (position, momentum) pairs (i.e., particles, as in our *dynamic with particles* model above) to parameterize $\theta$ indirectly.

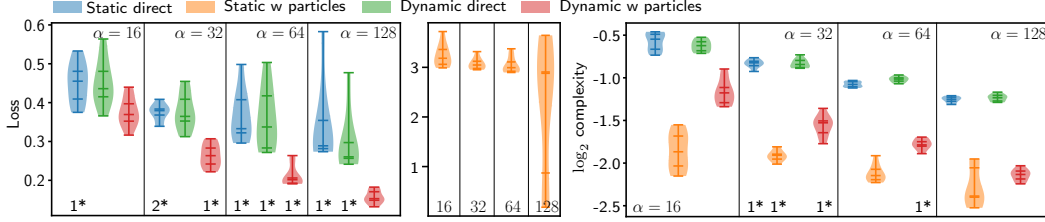

**Figure 2:** Fit for quadratic-like $y = x^2 + 3/(1 + x^2)$ for 10 random initializations. *Left*: Test loss; *Right*: time-integral of $\log_2$ of the Frobenius norm complexity. Lower is better for both measures. * indicates number of removed outliers (outside the interquartile range (IQR) by $\geq 1.5\times$ IQR); $\alpha$ denotes the inflation factor.

- Finally, we consider the **dynamic direct** model which uses a piece-wise time-constant $\theta(t)$. It essentially chains together multiple *static direct* models and is closely related to a discrete `ResNet` in the sense that multiple blocks (we use five) are used in succession. However, each block involves time-integrating the `UpDown` model. While the *dynamic with particles* model captures $\theta(t)$ indirectly via particles and shooting, the *dynamic direct* model requires many more parameters as it represents $\theta(t)$ directly. We show results for the *dynamic direct* model for a subset of the experiments.

All experiments use the `UpDown` model with quadratic penalty function $R$. Detailed experimental settings, including weights for the quadratic penalty function, can be found in Appendix F.

**Simple 1D function regression.** We approximate a simple quadratic-like function $y = x^2 + 3/(1 + x^2)$ which is non-invertible. We use 15 particles for our experiments. Fig. 2 shows the test loss and the network complexity, as measured by the log Frobenius norm integrated over time [28], for the different models as a function of the inflation factor $\alpha$ (cf. §3.2). On average, the *dynamic with particles* model shows the best fits with the lowest complexity measures, indicating the simplest network parameterization. Note that the *static with particles* approach results in the lowest complexity measures only because it cannot properly fit the function as indicated by the high test loss. Additional results for a cubic function $y = x^3$ are in Appendix G.

**Spiral.** Next, we revisit the spiral ODE example of [13] following the nonlinear dynamics $\dot{\mathbf{x}} = A\mathbf{x}^3$, $\mathbf{x} \in \mathbb{R}^2$ (where the power is component-wise). We fix $\mathbf{x}(0) = [2, 0]^T$, use $A = [-0.1, 2.0; -2, -0.1]$ and evolve the dynamics for time $T = 10$. The training data consists of snippets from this trajectory, all of the same length. We use an $L^2$ norm loss (calculated on all intermediate time-points) and 25 particles. Our goal is to show that we can obtain the best fit to the training data due to our dynamic model. Fig. 3 (*top*) shows that we can indeed obtain similar or better fits (lower losses) for a similar number of parameters while achieving the lowest network complexity measures. Fig. 3 (*bottom*) shows the corresponding results for the validation data consisting of the original long trajectory starting from initial value $\mathbf{x}(0)$. Interestingly, by pasting together short-range solutions we are successful in predicting the long-range trajectory despite training on short-range trajectory snippets.

**Concentric circles.** To study the impact of the inflation factor $\alpha$ in a classification regime, we replicate the concentric circles setting of [14]. The task is learning to separate points, sampled from two disjoint annuli in $\mathbb{R}^2$. While we are less interested in the learned flow (as in [14]), we study how often the proposed `UpDown` (dynamic with particles) model perfectly fits the training data as a function of $\alpha$. To the right, we show the success rate over 50 training runs for three choices of $\alpha$ and 20 particles. *Notably, the effect of $\alpha$ is only visible if the classification loss is down-weighted so that the regularization, R, dominates*. Otherwise, for the tested $\alpha$, the model always fits the data. The experiment is consistent with [14], where it is shown that increasing the space on which an ODE is solved allows for easy separation of the data and leads to less complex flows. The latter is also observed for our model.

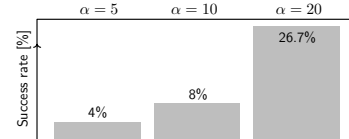

**Rotating MNIST.** Here, we are given sequences of a rotating MNIST digit (along 16 angles, linearly spaced in $[0, 2\pi]$). The task is learning to synthesize the digit at any rotation angle, given only the *first* image of a sequence. We replicate the setup of [40] and consider rotated versions of the digit "3". We identify each rotation angle as a time point $t_i$ and ran-

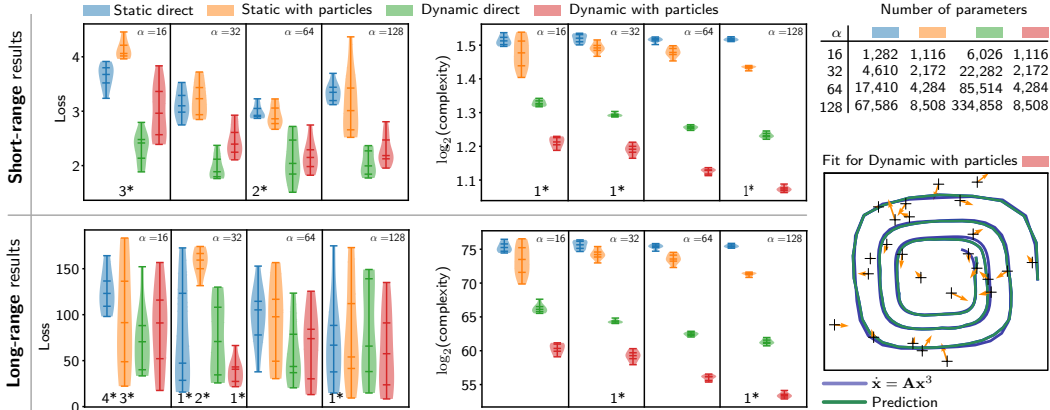

**Figure 3:** Fit for spiral (short- and long-range). Losses for the different models as well as the time-integral of $\log_2$ of the Frobenius norm complexity measure. Lower is better for both measures. The * symbol indicates how many outliers were removed and $\alpha$ denotes the inflation factor.

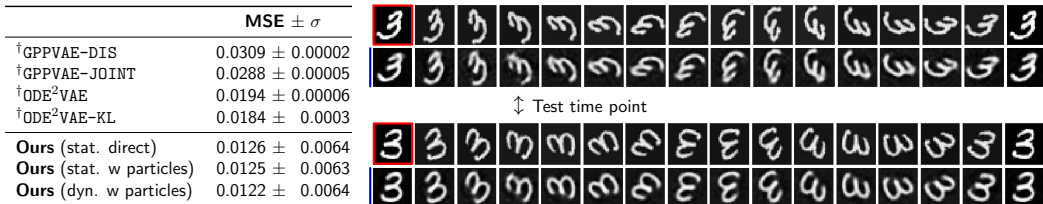

| | MSE $\pm \sigma$ |
|---|---|
| [†]GPPVAE-DIS | $0.0309 \pm 0.00002$ |
| [†]GPPVAE-JOINT | $0.0288 \pm 0.00005$ |
| [†]ODE$^2$VAE | $0.0194 \pm 0.00006$ |
| [†]ODE$^2$VAE-KL | $0.0184 \pm 0.0003$ |
| **Ours** (stat. direct) | $0.0126 \pm 0.0064$ |
| **Ours** (stat. w particles) | $0.0125 \pm 0.0063$ |
| **Ours** (dyn. w particles) | $0.0122 \pm 0.0064$ |

**Figure 4:** *Left*: Image (per-pixel) MSE (measured at the marked time point) averaged over all testing sequences of the rotated MNIST dataset. *Right*: Two testing sequences and predictions (marked blue) for all 16 time points when the image at $t = 0$ is given as input (marked red). Results marked with [†] are taken from [40].

domly drop four time points of each sequence during training. One fixed time point is consistently left-out and later evaluated during testing. We use the same convolutional autoencoder of [40] with the UpDown model operating in the internal representation space after the encoder. During training, the encoder receives the first image of a sequence (always at angle $0°$), the UpDown model integrates forward to the desired time points, and the decoder decodes these representations. As loss, we measure the mean-squared-error (MSE) of the decoder outputs. Fig. 4 lists the MSE (at the left-out angle), averaged over all testing sequences and shows two example sequences with predictions for all time points (100 particles, $\alpha = 10$).

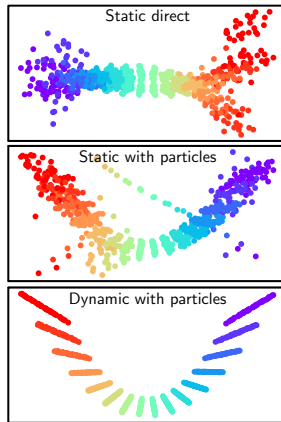

While all UpDown variants substantially lower the MSE previously reported in the literature, they exhibit comparable performance. To better understand the differences, we visualize the internal representation space of the autoencoder by projecting all 16 internal representations (i.e., the output of the UpDown models after receiving the output of the encoder) of each testing image onto the two largest principal components, shown to the right (different colors indicate the different rotation angles). This qualitative result shows that allowing for a time-dependent parameterization leads to a more structured latent space of the autoencoder.

**Bouncing balls.** Finally, we replicate the "bouncing balls" experiment of [40]. This is similar to the rotating MNIST experiment, but the underlying dynamics are more complex. In particular, we are given 10,000 (training) image sequences of bouncing balls at 20 different time points [34]. The task is learning to predict, after seeing the first three images of a sequence, future time points. We use the same convolutional autoencoder of [40] and minimize image (per-pixel) MSE (using all 20 time points for training). Our UpDown model operates in the internal representation space of the encoder (50-dimensional in our experiments[3]). In test mode, the network receives the first three image of a

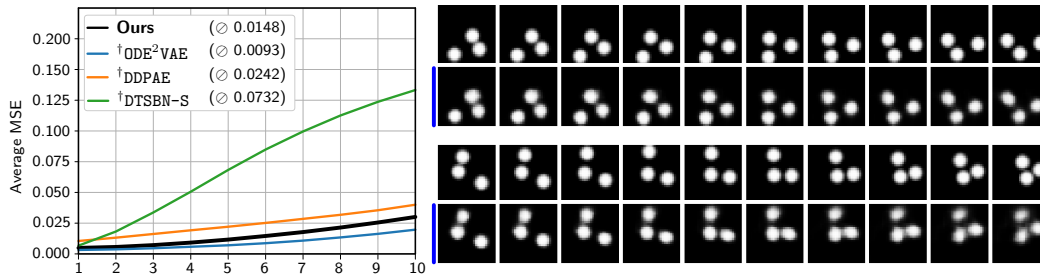

**Figure 5:** *Left*: Image (per-pixel) MSE for predicting 10 time points ahead (after receiving the first three inputs of a sequence), averaged over all testing sequences (numbers in parentheses indicate the MSE when additionally averaged over *all* prediction time points). Results marked with [†] are taken from [40]. *Right*: Two testing sequences with predictions (marked blue).

sequence and predicts 10 time points ahead. We measure the image (per-pixel) MSE and average the results (per time point) over all 500 testing sequences. For model selection, we rely on the provided validation set. Our `UpDown` (dynamic with particles) model uses 100 particles. Fig. 5 (*left*) lists the averaged MSE per time point, plotted against the approaches listed in [40]. Fig. 5 (*right*) shows two testing sequences with predictions (the three input time points are not shown). Results for the `UpDown` static and static with particles model are ⊘ 0.0154 and ⊘ 0.0150, respectively.

**Computational cost.** The computational cost of the `UpDown` model consists in storing the particles and running forward the model for the collection of particles and the data. Hence, computational cost scales linearly in the number of particles. To get rid of this linear relationship (in case only a forward pass is needed), the ODE can be discretized in time and the `ResNet` with its weights is obtained.

## 5 Discussion and Conclusions

We demonstrated that it is possible to parameterize DNNs via initial conditions of (position, momentum) pairs. While our experiments are admittedly still simple, results are encouraging as they show that 1) the particle-based approach can achieve competitive performance over direct parameterizations and that 2) time-dependent parameterizations are useful for obtaining simpler networks and can be realized with significantly fewer parameters using particle-based shooting.

Our work opens up many different follow-up questions and formulations. For example, we presented our approach for a model with continuous dynamics, but the particle and the shooting formalism can also be applied to discrete-time models. Further, we focused, for simplicity, on continuous variants of multi-layer perceptrons, but similar linear-in-parameter models can be formulated for convolutional neural networks. Models that are nonlinear in their parameters hold the promise for connections with optimal mass transport theory and to theoretical complexity results, which we touched upon for our `UpDown` model. Indeed, this change of paradigm in the parameterization may result in new quantitative results on network generalization properties. Lastly, how well the approach generalizes to more complex problems, how many particles are needed to switch from a standard deep network to its shooting formulation, and how optimizing over critical points of the original optimization problem via shooting relates to network generalization will be fascinating to explore.

**Source code** is available at: `https://github.com/uncbiag/neuro_shooting`

## Broader Impact

One goal of this work is to enrich the understanding of continuous depth neural networks and to open a different (or alternative) perspective on its parameterization. Specifically, we shift the parameterization of deep neural networks from a layer-by-layer perspective to an initial-value perspective and Hamiltonian dynamics. At this point, our work is conceptual and theoretical in nature; broader impact emerges most likely as a consequence of better understanding the role of neural network parameterizations.

## Acknowledgments and Disclosure of Funding

This research project was initiated during a one-month invitation of M. Niethammer by the Labex Bézout, supported by the French National Research Agency ANR-10-LABX-58. R. Kwitt is partially funded by the Austrian Science Fund (FWF): project FWF P31799-N38 and the Land Salzburg (WISS 2025) under project numbers 20102-F1901166-KZP and 20204-WISS/225/197-2019. S. Wei is the recipient of an Australian Research Council Discovery Early Career Award (project number DE200101253) funded by the Australian Government.

## Footnotes

[1]In a discrete setting, resolving the forward model in the backward direction generally requires costly solving of implicit equations. This can be done (it is, e.g., done for invertible `ResNets` [8]). In general, an explicit numerical solution for forward time-integration becomes implicit in the backward direction and vice versa.

[2]Obviously, an affine function of the parameters also works similarly.

[3]We did not further experiment with this hyperparameter, so potentially better results can be obtained.

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
