[Supplementary Material]

## Supplementary material

The following sections discuss in more detail the theoretical guarantees of our approach. §A presents the optimality conditions underlying our shooting formulation and it is shown how these optimality equations can be approximated via a collection of particles. §B proposes different regularizations, whose choice is key for practical and theoretical results. We show that, under some conditions, the Rademacher complexity of the set of flows can be bounded and apply our results in §B.5 to the UpDown model. §C discusses the number of free parameters of our shooting approach in relation to the number of free parameters for direct optimization. §D explains how the shooting equations can be automatically derived via automatic differentiation. §E shows the universality of our UpDown model. §F provides details on our experimental setup. Lastly, §G shows some additional experimental results.

## A    Expectation approximation of optimality equations

We first discuss a general variational setup of supervised learning including regularization.

### A.1    Variational setup

Suppose the data consists of input $X \in \mathbb{R}^d$. Let $f(\cdot, \theta(t))$ be a vector field on $\mathbb{R}^d$, e.g. the single hidden layer of Eq. (3.7) or a linear (in parameter) layer. Consider the flow $\varphi := \varphi(T, \cdot)$ generated by $f$ according to

$$
\begin{cases}
\frac{\mathrm{d}}{\mathrm{d}t} \varphi(t, \mathbf{x}) = f(\varphi(t, \mathbf{x}), \theta(t)) \,, \\
\varphi(0, \mathbf{x}) = \mathbf{x} \,.
\end{cases}
\tag{A.1}
$$

We consider the general task of minimizing,

$$
\mathrm{Reg}(\varphi) + \gamma \mathbb{E}[\ell(\varphi(X))] \,,
\tag{A.2}
$$

where $\gamma$ is a positive regularization parameter.

We now consider the particular case of a ResNet model where each layer is given by an UpDown model Eq. (3.9) or even a single hidden layer Eq. (3.7).

Without loss of generality, set the terminal time to $T = 1$. Letting $\rho_0$ denote the probability density of $X$, minimizing Eq. (A.2) is equivalent to minimizing

$$
\inf_{\varphi} \left[ \mathrm{Reg}(\varphi) + \gamma \int_{\mathbb{R}^d} \ell(\varphi(\mathbf{x})) \rho_0(\mathbf{x}) \, \mathrm{d}\mathbf{x} \right] \quad .
$$

This can be rewritten as

$$
\inf_{\varphi} \left[ \mathrm{Reg}(\varphi) + \gamma \int_{\mathbb{R}^d} \ell(\mathbf{x}') \rho_1(\mathbf{x}') \, \mathrm{d}\mathbf{x}' \right] \quad ,
$$

where $\rho_1(\mathbf{x}) := \rho(1, \mathbf{x})$ is the flow of the continuity equation

$$
\partial_t \rho(t, \mathbf{x}) + \mathrm{div}(\rho(t, \mathbf{x}) f(\mathbf{x}, \theta)) = 0 \,, \rho(0, \mathbf{x}) = \rho_0(\mathbf{x}) \,,
$$

where $\mathrm{div}$ is the divergence operator on vector fields. Note that $\rho_1$ can be regarded as the density representing the data at time 1. In the following, we deal with a general regularization term $\mathrm{Reg}(\varphi) = \int_0^1 R(\theta(t), \rho(t)) \, \mathrm{d}t$, where the $R$ term can depend on the density of data at time $t$. A particular though important case is when the regularization $R$ does not depend on $\rho$, $\int_0^1 R(\theta(t)) \, \mathrm{d}t$.

### A.2    Optimality equations and Hamiltonian ensemble approximation

We detail the optimality equations when data points are represented by a probability measure. As mentioned above, the regularity of the map is enforced via a penalty on the weights at each timepoint and is the integral $\int_0^1 R(\theta(t)) \, \mathrm{d}t$ or even more generally $\int_0^1 R(\theta(t), \rho(t)) \, \mathrm{d}t$. Using Lagrange multipliers, this constraint can be enforced and minimizers of the energy should be saddlepoints of the

energy

$$\mathcal{L}(\rho, \theta, p) := \gamma \int_{\mathbb{R}^d} \ell(\mathbf{x}) \rho_1(\mathbf{x}) \, \mathrm{d}\mathbf{x} + \int_0^1 R(\theta(t), \rho(t)) \, \mathrm{d}t$$
$$+ \int_0^1 \int_{\mathbb{R}^d} p(t, \mathbf{x}) (\partial_t \rho(t, \mathbf{x}) + \mathrm{div}(\rho(t, \mathbf{x}) f(\mathbf{x}, \theta(t)))) \, \mathrm{d}\mathbf{x} \, \mathrm{d}t \,,$$

where $p(t, \mathbf{x})$ is a time and space dependent function. The optimality equations are then

$$\begin{cases} \partial_t \rho(t, \mathbf{x}) + \mathrm{div}(\rho(t, \mathbf{x}) f(t, \mathbf{x}, \theta(t))) = 0 \,, \\ \partial_t p(t, \mathbf{x}) + \nabla p(t, \mathbf{x}) \cdot f(t, \mathbf{x}, \theta(t)) = \frac{\delta R}{\delta \rho}(\theta(t), \rho(t)) \,, \\ \partial_\theta R(\theta(t), \rho(t)) - \int_{\mathbb{R}^d} \partial_\theta f(\mathbf{x}, \theta(t))^\top (\nabla p(t, \mathbf{x}) \rho(t, \mathbf{x})) = 0 \,, \end{cases} \tag{A.3}$$

where $\nabla p$ is the gradient w.r.t. $\mathbf{x}$ of $p(t, \mathbf{x})$ and $\delta$ denotes differentiation w.r.t. the indicated parameter. The notation $\frac{\delta R}{\delta \rho}$ means the Fréchet derivative of the penalty w.r.t. the density $\rho$. Note that in our current work, $R$ is independent of $\rho$. However, this more general setup encompasses optimal transport models, see Section B.2.

In practice, one does not have access to the full distribution and the variational setup needs to be approximated. As proposed in the main text, we approximate it using a collection of particles that follow the optimality equations which are Hamiltonian evolution equations for this collection of particles. The collection of particles $\{(\mathbf{q}_j, \mathbf{p}_j)\}$ are defined by their state and costate. We estimate $\rho$ using the empirical measure $\frac{1}{K} \sum_{j=1}^K \delta_{\mathbf{q}_j(t)}(\cdot)$. Writing the optimality equation for this particular empirical measure leads to the equation Eq. (3.4). When the number of particles tends to infinity, we can hope to recover the optimal trajectory. However, we do not explore this question formally here. We simply remark that this question is directly connected to expressiveness and generalization properties of the constructed neural network and is also probably data dependent.

# B   Choice of regularization

The simplest regularization on the flow $\varphi$ is given by

$$\mathrm{Reg}(\varphi) = \int_0^1 R(\theta(t)) \, \mathrm{d}t \,, \tag{B.1}$$

where $R$ does not depend on $\rho(t, \cdot)$. The first possibility is a quadratic penalty for the single-hidden-layer vector field of Eq. (3.7), where $R(\theta(t)) = \frac{1}{2} \|\theta(t)\|_2^2$ is the Frobenius norm of the parameter $\theta$. Since the space of vector fields is a *finite* dimensional linear space, it can be endowed with a scalar product, which turns this space into a Reproducing Kernel Hilbert Space (RKHS). Therefore, the linear in parameter - quadratic penalty setting of §3.1 is a particular case of vector fields encoded by $f(\cdot, \theta(t)) \in H$, with $H$ a RKHS embedded in $W^{1,\infty}$ vector fields. This setting leverages strong analytical and geometrical foundations [41, 11]:

1) When the activation function is smooth, the resulting vector field is smooth[4], and consequently the associated flow map $\varphi$ is guaranteed to be a one-to-one smooth map (i.e., a diffeomorphism). For instance, with the UpDown model, it is a homeomorphism in $(\mathbf{x}, \mathbf{v})$. Moreover, the quadratic penalty induces a right-invariant distance on the set of flows generated by Eq. (A.1) and the distance to identity of the resulting flow can be bounded by $\mathrm{Reg}(\varphi)$ (see [41, 11] for more details in a Sobolev setting). 2) When the activation function is of ReLU type, the resulting map is still a $W^{1,\infty}$ one-to-one map (i.e., a homeomorphism) and has Lipschitz regularity.

Another type of regularization for the single-hidden-layer vector field of Eq. (3.7) we discuss is based on the Barron norm [15]:

$$\|\theta\|_{\mathcal{B}}^2 := \frac{1}{d'} \sum_{j=1}^{d'} \|\theta_1^j\|_2^2 (\|[\theta_2]_j\|_1 + \|b_2^j\|_1)^2 \,,$$

where $\theta_1^j$ denotes the $j^{\text{th}}$ column of $\theta_1$ and $[\theta_2]_j$ denotes the $j^{\text{th}}$ row of $\theta_2$. As discussed in the main text, the reason we might consider a Barron norm penalty for the single-hidden-layer vector field in

Eq. (3.7) rather than the quadratic penalty is because of its theoretical results. Indeed, the Rademacher complexity is bounded for the combination of a single-hidden-layer vector field with a Barron norm penalty, but not when combined with a quadratic penalty.

## B.1 Linear in parameters - quadratic energy

Now let us examine in detail models that are *linear* in parameters and have *quadratic* energy on parameters: this case is the simplest to be studied, and computationally not as demanding as the nonlinear case. As mentioned above, the set of possible vector fields $f(\cdot, \theta(t))$ is a finite dimensional linear space, which is a reproducing kernel Hilbert space when endowed with an $L^2$ norm. Since all Hilbert norms in finite dimensions are equivalent, this choice of regularization is universal in this class of quadratic penalties.

1. The vector field is $f(\cdot, \theta(t)) = \theta \cdot \sigma$, where $\sigma$ is a vector of maps. In this case, the optimality equation reads

$$\partial_\theta f(\mathbf{x}, \theta(t))^\top (\nabla p(t, \mathbf{x})\rho(t, \mathbf{x})) = \int_{\mathbb{R}^d} \sigma(x)^\top (\nabla p(t, \mathbf{x})\rho(t, \mathbf{x})) \, \mathrm{d}\mathbf{x}.$$

2. If the penalty $R$ only depends on $\theta$ and is quadratic: $R(\theta(t)) = \frac{1}{2} \int_0^1 \|\theta(t)\|^2 \, \mathrm{d}t$, then one has $\frac{\delta R}{\delta \theta}(\theta(t), \rho(t)) = \theta(t)$.

Thus, under these two conditions, the parameters are *explicit* in terms of $p$, $\rho$ and $\sigma$:

$$\theta(t) = \int_{\mathbb{R}^d} \sigma(\mathbf{x})^\top (\nabla p(t, \mathbf{x})\rho(t, \mathbf{x})) \, \mathrm{d}\mathbf{x}. \tag{B.2}$$

Two observations are warranted. First, if, instead of quadratic regularization on the parameters, we were to choose a RKHS norm (in the infinite dimensional case) as penalty, it would result in the introduction of the kernel applied to the R.H.S. of Eq. (B.2). Second, from Eq. (B.2), one could be tempted to derive an evolution equation for $\theta$. This equation is known as the EPDiff equation [41] and is unfortunately not a closed equation on the set of parameters $\theta(t)$ themselves. Therefore, our approach is a possible way to approximate it.

An important property of this simple setting is that the norm of the vector field is preserved by the forward model defined by the collection of Hamiltonian particles and it also holds in the continuous setting. As stated in Section 3.1, the Hamiltonian is given by $R(\theta(t)) = \frac{1}{2} \operatorname{Tr}\left(A(t)^\top M_A A(t)\right) + \frac{1}{2} b(t)^\top M_b b(t)$ where $A, b$ are the optimal parameters given by

$$\begin{cases} A(t) &= M_A^{-1}(-\sum_{j=1}^K \mathbf{p}_j(t)\sigma(\mathbf{q}_j(t))^\top) \\ b(t) &= M_b^{-1}(-\sum_{j=1}^K \mathbf{p}_j(t)). \end{cases} \tag{B.3}$$

The Hamiltonian $R(\theta(t))$ being constant gives a constant norm vector field.

## B.2 Nonlinear in parameters - energy which depends on the distribution

For exposition purposes, we present two cases of interest which we have not well explored numerically.

**Example of the Barron norm.** Obviously, the single-hidden-layer vector field in Eq. (3.7) is not linear in parameters. We have already discussed that it is proper in this case to endow the space with norms such as the Barron norm [15]. For simplicity, consider the single-hidden-layer vector field in Eq. (3.7) without $b_1$, i.e., $f(\mathbf{x}(t), \theta(t)) = \theta_1 \sigma(\theta_2(\mathbf{x}) + b_2)$. A simple upper bound for the Barron norm[5] is

$$\|f(\cdot, \theta)\|_{\mathcal{B}}^2 := \frac{1}{d'} \sum_{j=1}^{d'} \|\theta_1^j\|_2^2 (\|[\theta_2]_j\|_1 + \|[b_2]_j\|_1)^2. \tag{B.4}$$

Again, $\theta_1^j$ denotes the $j^{\text{th}}$ column of $\theta_1$ and $[\theta_2]_j$ denotes the $j^{\text{th}}$ row of $\theta_2$.

Let us consider the case of $R(\theta(t)) = \frac{1}{2}\|f(\cdot,\theta)\|_{\mathcal{B}}^2$. In this case, one has the following optimality equations to solve

$$\theta_1^j(\|[\theta_2]_j\|_1 + \|\|[b_2]_j\|_1)^2 = \int_{\mathbb{R}^d} \sigma([\theta_2]_j\mathbf{x} + [b_2]_j)^\top (\nabla p(t,\mathbf{x})\rho(t,\mathbf{x}))\,\mathrm{d}\mathbf{x}\,,$$

$$\|\theta_1^j\|_2^2(\|[\theta_2]_j\|_1 + \|[b_2]_j\|_1)\partial\|[\theta_2]_j^k\|_1 = \int_{\mathbb{R}^d}[\mathrm{d}\sigma([\theta_2]_j\mathbf{x} + [b_2]_j)(\mathbf{x}_k)]^\top (\nabla p(t,x)\rho(t,\mathbf{x}))\,\mathrm{d}\mathbf{x}\,,$$

$$\|\theta_1^j\|_2^2(\|[\theta_2]_j\|_1 + \|[b_2]_j\|_1)\partial\|[b_2]_k^j\|_1 = \int_{\mathbb{R}^d}[\mathrm{d}\sigma([\theta_2]_j\mathbf{x} + [b_2]_j)(\mathbf{x}_k)]^\top (\nabla p(t,\mathbf{x})\rho(t,\mathbf{x}))\,\mathrm{d}\mathbf{x}\,.$$

These equations involve the subdifferential of the $L^1$ norm, and optimization of this type of functions, which involves sparsity, is a well-explored field [5]. We leave experiments with this norm for future work. Note that in this case the norm of the vector field is not equal to the Hamiltonian and it is not a constant of the flow.

## B.3 $L^2$ regularization, optimal transport

Last, we briefly mention a model that is part of our framework which has the advantage of not specifying the penalty on the space of parameters encoding the vector field. In case there is no obvious norm to be used on the space of vector fields, it is possible to use an $L^2$ type of penalty on the vector fields themselves instead of on the parameters.

Indeed, one way to be rather independent of the choice of the parameterization of the map consists in introducing a cost that represents the $L^2$ norm of the map. However, $L^2$ depends on the choice of a measure and this measure can be chosen as the density of the data, $\rho(t,\mathbf{x})$. More precisely, one can use

$$R(f,\rho(t)) = \frac{1}{2}\int_{\mathbb{R}^d}\|f(\mathbf{x},\theta)\|^2\rho(t,\mathbf{x})\,\mathrm{d}\mathbf{x}\,. \tag{B.5}$$

In such a case, this formulation resembles finding an optimal transport (OT) map between $\rho_0$ and $\rho_1$. Specifically, optimal transport is an optimization problem which can be solved via a fluid dynamic formulation [9] introducing the kinetic penalty above. However, the two models (OT and the one defined by the regularization Eq. (B.5)) differ since the optimization set for optimal transport is the set of $L^2$ vector fields with respect to measure $\rho$ and the above formulation is a parameterized approximation of this set.

This parameterized approximation needs to retain generalization properties of the optimized map. Note however, that in the limit where the number of neurons goes to infinity, optimal transport will be well-approximated since the optimization is performed on a dense subset of all vector fields. Obviously, fixing the choice to a single-hidden-layer design implies a choice for $d'$ in $\theta_1(t) \in L(\mathbb{R}^{d'},\mathbb{R}^d)$ and $\theta_2(t) \in L(\mathbb{R}^d,\mathbb{R}^{d'})$ of Eq. (3.7), which thus gives a regularization of the computed approximation of the optimal transport map.

**Computational burden.** In either case of the Barron norm or the optimal transport type of penalty, the implicit equation corresponding to the third equation in Eq. (A.3) has to be solved at each layer of the discretization. We experimented with a simple strategy of unrolling the related minimization scheme. An efficient approach to solve such implicit equations will be necessary for practical implementations.

## B.4 Rademacher complexity of bounded energy flows.

In this section, given a set of vector fields with bounded Rademacher complexity, we show that the resulting flows also have bounded Rademacher complexity. The flow of a vector field $f(\cdot,\theta(t))$ is a vector valued map denoted by $\varphi$. Let us first treat the case of the Rademacher complexity of a component of the flow map $\varphi^k$.

**Theorem 2.** *Let $\mathcal{F}$ be a space of vector fields defined on a compact space $C \subset \mathbb{R}^d$. Assume that the Rademacher complexity on $n$ points in $C$ of each component of the vector fields $f^k(t,\cdot)$ for $k = 1,\ldots,d$ is controlled by $M(n,t)$ which depends on $n$, then the Rademacher complexity of each component of the flows at time $1$ is bounded by $\int_0^1 M(n,t)\,\mathrm{d}t$.*

*Proof.* Recall that Rademacher complexity, see [39], of a class of functions $\mathcal{F}$ is defined as, for $\mathbf{Z} = (\mathbf{z}_1, \ldots, \mathbf{z}_n) \in C$,

$$\mathrm{Rad}_{\mathbf{Z}}(\mathcal{F}) \stackrel{\mathrm{def.}}{=} \mathbb{E}\left[\sup_{g \in \mathcal{F}} \sum_{i=1}^n \varepsilon_i g(\mathbf{z}_i)\right],$$

where the $\{\varepsilon_i\}_{i=1}^n$ are i.i.d. Rademacher random variables. Our hypothesis ensures $\mathrm{Rad}_{\mathbf{Z}}(\mathcal{F}) \leq M(n)$. Apply the definition of the flow to get

$$\varphi(1, \mathbf{x}) = \mathbf{x} + \int_0^1 f(\varphi(t, \mathbf{x}), \theta(t)) \, \mathrm{d}t.$$

Therefore,

$$\mathbb{E}\left[\sup_{\varphi \in \mathcal{F}} \sum_{i=1}^n \varepsilon_i \varphi^k(\mathbf{z}_i)\right] \leq \mathrm{Rad}_{\mathbf{Z}}(\{\mathrm{Id}\}) + \int_0^1 \mathbb{E}\left[\sup_{f(\cdot, \theta(t))} \sum_{i=1}^n \varepsilon_i f^k(\varphi(t, \mathbf{z}_i), \theta(t))\right] \mathrm{d}t,$$

$$\leq 0 + \int_0^1 M(n, t) \, \mathrm{d}t.$$

In the previous formula, we used the fact that the Rademacher complexity of a set comprised of a single map is zero. □

**Corollary 3.** *Let $H$ be a RKHS of vector fields whose kernel $\mathsf{k}$ is bounded on the diagonal $\|\mathsf{k}(\mathbf{x}, \mathbf{x})^k\|_\infty < \infty$, then, the set of flows denoted by $\mathcal{F}$ at time 1 of time-dependent vector fields in $\mathcal{B}(0, R)$, the ball of radius $R$ centered at the origin satisfies $\mathrm{Rad}_{\mathbf{Z}}(\mathcal{F}) \leq \frac{2R\sqrt{\|\mathsf{k}(\mathbf{x}, \mathbf{x})^k\|_\infty}}{\sqrt{n}}$, where $\mathrm{Rad}_{\mathbf{Z}}(\mathcal{F})$ is the Rademacher complexity for $n$ points.*

*Proof.* The Rademacher complexity of the ball of radius $R$ in the RKHS $H$ [7, Lemma 22] is upper bounded: $\mathrm{Rad}_{\mathbf{Z}}(\mathcal{B}(0, R)) \leq \frac{2R\sqrt{\|\mathsf{k}(\mathbf{x}, \mathbf{x})^k\|_\infty}}{\sqrt{n}}$. We then directly apply Theorem 2. □

A similar result also holds for vector fields generated by the single-hidden-layer vector field in Eq. (3.7), see [15]. Last, we note that the result and its proof also hold if one uses the following Rademacher complexity for vector valued functions [26],

$$\mathrm{Rad}_{\mathbf{Z}}(\mathcal{F}) \stackrel{\mathrm{def.}}{=} \mathbb{E}\left[\sup_{g \in \mathcal{F}} \sum_{i=1}^n \sum_{j=1}^d \varepsilon_j \varepsilon_i g_j(\mathbf{z}_i)\right],$$

for $g = (g_j)_{j=1,\ldots,d} \in \mathcal{F}$.

### B.5 Consequences for the `UpDown` model

We put together the previous results on the `UpDown` model. First, the space of vector fields endowed with the quadratic penalty on the parameters forms a RKHS. The variational formulation implies that the norm of the velocity field generated by a given collection of Hamiltonian particles $\{(\mathbf{q}_j(0), \mathbf{p}_j(0))\}$ is preserved. In addition, this norm can be explicitly computed since the parameters at time 0 can be computed in terms of $\{(\mathbf{q}_j(0), \mathbf{p}_j(0))\}$. Last, the generated space of maps has a Rademacher complexity which is linearly bounded by this norm. In order to be fully explicit on the constant for the Rademacher complexity, we need to compute $\sup_{x \in C} \|\mathsf{k}(x, x)\|$ where $\mathsf{k}$ is the kernel associated with the RKHS. Without making this quantity explicit here, we simply mention that the bound degrades (i.e. increases) with increasing inflation factor $\alpha$, as it can be expected.

## C  Analysis of the number of free parameters

It is instructive to understand the number of parameters for a shooting approach in comparison to the typical approach of optimizing a neural network (where the parameter-dependency at optimality is only considered implicitly at convergence of the numerical solution rather than explicitly during the shooting). We focus on the cases of affine and convolutional layers for illustration.

Consider a DNN with a depth of $L$ layers, where each hidden layer has $P$ parameters. The number of free parameters is then $LP$, compared to $2KS$ where $K$ is the number of active particles, each of them of size $2S$[6]. Hence, solutions with less than $LP/(2S)$ particles provide benefits in the number of free parameters. Therefore, as the number of particles is reduced, we may parameterize the DNN with a smaller number of parameters. *Most remarkably, the number of free parameters is always $2KS$ regardless of the number of parameters of a particular layer as the layer parameters are obtained via the shooting equations based on the particle states.* This is a consequence of regularizing the parameters in our loss which couples them across time at optimality. We make this clearer in what follows.

**Affine layers.** Recall that in our simple example of §3.1 the parameters $\theta(t) = [A(t), b(t)]$ of our affine[7] model are given as

$$A(t) = M_A^{-1}\left(-\sum_{j=1}^{K} \mathbf{p}_j(t)\sigma(\mathbf{q}_j(t))^\top\right), \quad b(t) = M_b^{-1}\left(-\sum_{j=1}^{K} \mathbf{p}_j(t)\right) \ . \quad \text{(C.1)}$$

Here, $A(t)$ and $b(t)$ have $d^2$ and $d$ parameters, respectively; these parameters are indirectly given by the set of particles $\{(\mathbf{q}_j(t), \mathbf{p}_j(t))\}$ at any given time. Hence, for this model $S = d$ and $P = d(d+1)$. If we assume we have $K$ particles and compare to a discrete layer implementation of this model then the particle-based approach will have less free parameters if

$$2Kd < Ld(d+1) \ .$$

Importantly, the state-space dimension, $d$, only enters the number of free parameters linearly for the particle approach ($2Kd$), while there is a quadratic dependence for direct optimization ($Ld(d+1)$). This is a direct consequence of the optimality condition which couples the parameters $\theta(t)$ across time. One can see this phenomenon in action in Eq. (C.1), where the matrix $A$ is expressed as the sum of matrices $\mathbf{p}_j(t)\sigma(\mathbf{q}_j(t))^\top$ with rank $\leq 1$. Concretely, a particle-based shooting approach uses less parameters if the number of particles $K < L(d+1)/2$. Another interesting observation based on this example is that even if we would have only considered a linear model (i.e., without the bias term, $b(t)$) the number of parameters for the particles would have still remained at $2KS$. This is again a consequence of optimality and of our parameter regularization. Note that this also means that even though our UpDown model

$$\dot{\mathbf{x}}(t) = \theta_1(t)\sigma(\mathbf{v}(t)) + b_1(t), \quad \dot{\mathbf{v}}(t) = \theta_2(t)\mathbf{x}(t) + b_2(t) + \theta_3(t)\sigma(\mathbf{v}(t)) \,,$$

has significantly more parameters $\theta(t) = [\theta_1(t), b_1(t), \theta_2(t), b_2(t), \theta_3(t)]$ when directly optimized, this has no direct impact on the number of free parameters of its particle-based parameterization. Only the state-space dimension matters. Concretely, if we were to instead consider a model of the form

$$\dot{\mathbf{x}}(t) = \theta_1(t)\sigma(\mathbf{v}(t)), \quad \dot{\mathbf{v}}(t) = \theta_2(t)\mathbf{x}(t) \,,$$

the particle-based parameterization would stay *unchanged!* Only the way how one infers $\theta(t)$ from the particles changes.

**Convolutional layers.** Shooting approaches for convolutional models can also be derived. We did not experiment with such models in this work. However, we show here that the number of free parameters may also be decreased with a particle-based approach. This will be interesting to explore in future work. Specifically, for convolutional layers a particle-based parameterization could be particularly effective as one typically has quadratic complexity in the number of filters between convolutional layers (i.e., if a layer with $N$ feature channels is followed by a layer with $M$ feature channels, this will induce the estimation of $N \times M$ convolutional filters and hence will drastically influence the number of parameters for large $N$ or $M$). In contrast, a particle-based shooting approach

does not increase the number of parameters as it ties them together via the optimality conditions expressed by the shooting equations. As a rough estimate for a standard convolutional `ResNet` for $L = 50$, $P = 100^2 \times 16$, $LP \approx 8.10^6$. Thus, if particles have size 40, we end up with at most $10^5$ active particles.

**General remarks.** Nevertheless, all model parameters (e.g., $[A(t), b(t)]$ or all convolutional filters for a convolutional layer) are still instantiated during computation. It is important to note that regardless of the chosen number of particles, a shooting neural network solution is a possible optimal solution (for a given data set) at any given time, not only at convergence. One optimizes over the family of possible neural network models with the goal of finding the element within this family that best matches the observations.

## D    Automatic shooting

The general shooting equations were presented in Eq. (3.3). We then proceeded to explicitly derive the shooting equations for a continuous DNN with linear-in-parameter layers and `UpDown` layers in §3.1 and §3.2, respectively. While this was instructive, it is somewhat cumbersome, in particular, for more complex models or when moving to convolutional networks. Fortunately, in practice these shooting equations do not need to be derived by hand. Indeed, they are completely specified by the Hamiltonian

$$H(\mathbf{p}, \mathbf{x}, \theta) = \mathbf{p}^\top (\dot{\mathbf{x}} - f(t, \mathbf{x}, \theta)) + R(\theta) \ ,$$

in the sense that the shooting equations in Eq. (3.3) are computed via differentiation of $H$. Specifically, the shooting equations in Eq. (3.3) are equivalently given by

$$\begin{cases} \dot{\mathbf{x}} = \frac{\partial H(\mathbf{p}, \mathbf{x}, \theta)}{\partial p} \,, \\ \dot{\mathbf{p}} = -\frac{\partial H(\mathbf{p}, \mathbf{x}, \theta)}{\partial x} \,, \\ \theta \in \arg\min_\theta H(\mathbf{p}, \mathbf{x}, \theta) \,. \end{cases}$$

As discussed above, the last equation can be replaced by solving

$$\partial_\theta R(\theta) - \sum_{i=1}^{N} \partial_\theta f(t, \mathbf{x}_i, \theta)^T (\mathbf{p}_i) = 0 \ .$$

Automatic differentiation can be used to automatically obtain the shooting equations. As fitting a shooting model requires differentiating the shooting equations, we in effect end up with differentiating twice. This can be done seamlessly using modern deep learning libraries, such as `PyTorch`.

## E    Universality of the `UpDown` model

In this section, we set out to demonstrate that the `UpDown` model is universal in the sense that its associated flow can come $\varepsilon$-close to the flow of any well behaving time-dependent vector field.

Recall the single-hidden-layer vector field in Eq. (3.7) with time-varying parameters $\theta(t) = (\theta_1(t), \theta_2(t), b_1(t), b_2(t))$. While shooting with the single hidden layer vector field is theoretically appealing as it is universal [1], it would result in implicit shooting equations. We first show that the `UpDown` model introduced in 3.2 can give the same flow as the single hidden layer (Lemma 5) and then leverage this relationship to show that the `UpDown` model inherits the universality of the single hidden layer (Proposition 6).

**Lemma 4.** *Consider the single-hidden-layer vector field in Eq. (3.7) with $\theta_2(t)$ and $b_2(t)$ being piecewise $C^1$ and $\theta_1(t), b_1(t)$ continuous. Then, there exists a parameterization of the `UpDown` model that gives the same flow at a fixed time, $T = 1$.*

*Proof.* We rewrite the differential equation

$$\dot{\mathbf{q}}(t) = \theta_1(t)\sigma(\theta_2(t)\mathbf{q} + b_2(t)) + b_1(t) \,,$$

by introducing the additional state variable $\mathbf{v}(t) = \theta_2(t)\mathbf{q}(t) + b_2(t)$ which we differentiate w.r.t. time. We obtain $\dot{\mathbf{v}}(t) = \dot{\theta}_2(t)\mathbf{q}(t) + \dot{b}_2(t) + \theta_2(t)\dot{\mathbf{q}}(t)$. Replacing $\dot{\mathbf{q}}(t)$ by its formula, we get

$$\dot{\mathbf{v}}(t) = \dot{\theta}_2(t)\mathbf{q}(t) + \dot{b}_2(t) + \theta_2(t)\theta_1(t)(\sigma(\mathbf{v}(t)) + b_1(t)) \,.$$

The system can be rewritten as

$$\begin{cases} \dot{\mathbf{q}}(t) = \theta_1(t)\sigma(\mathbf{v}(t)) + b_1(t) \,, \\ \dot{\mathbf{v}}(t) = \theta_3(t)\mathbf{q}(t) + \theta_4(t)\sigma(\mathbf{v}(t)) + b_3(t) \,. \end{cases} \tag{E.1}$$

Therefore, with the initial condition $\mathbf{v}(0) = \theta_2(0)\mathbf{q}(0)$ and $\mathbf{q}(0) = \mathbf{q}_0$, the two systems of ordinary differential equations are equivalent. □

Note that the key point in Lemma 4 is the loss of regularity in the evolution of $\theta_2$ since we differentiated once in time. For that reason, we now show that adding more dimensions using the inflation factor $\alpha$ alleviate this issue. It is likely possible that one could prove a universality result using only $\alpha = 1$ but we shall leave this question for future work[8]. However, experimentally, the inflation factor has a crucial effect on the performance of the optimization, as discussed in §4. Lemma 4 helps us establish the next result.

**Lemma 5.** *Consider the single-hidden-layer vector field in Eq. (3.7) with $\theta(t)$ being piecewise continuous. Then, there exists a parameterization of the* UpDown *model that gives the same flow.*

*Proof.* Without loss of generality, we only treat the case of one discontinuity in time of the parameterization; We thus assume that $\theta(t)$ is continuous on $[0, t_1[$ and $[t_1, 1]$. We consider $\mathbf{q}, \mathbf{v}_1, \mathbf{v}_2 \in \mathbb{R}^d$ such that $\mathbf{q}, \mathbf{v}_1$ are defined as in Lemma 4. We now define, up to time $t_1$, $\mathbf{v}_2(t) = \theta(t_1)\mathbf{v}_1(t) + \theta_2(t_1)$ which implies (differentiating w.r.t. time) that $\mathbf{v}_2$ follows an evolution equation similar to $\mathbf{v}_1$ and thus can be encoded in the general form of Eq. (E.1). Now, $\mathbf{q}(t), \mathbf{v}_2(t)$ are defined on $[t_1, 1]$ by the evolution Eq. (E.1) in order to coincide with the flow of single-hidden-layer vector field on $[t_1, 1]$, $\dot{\mathbf{q}}(t) = \theta_1(t)\sigma(\mathbf{v}_2(t)) + b_1(t)$ and $\dot{\mathbf{v}}_2(t) = \theta_3(t)\mathbf{q}(t) + \theta_4(t)\sigma(\mathbf{v}_2(t)) + b_3(t)$ for well chosen parameters as in Lemma 4. Since the value of $\mathbf{v}_1(t)$ is not used in the evolution equation of $\mathbf{q}(t)$, we can simply extend it by $\mathbf{v}_1(t) = \mathbf{v}_1(t_1)$ which is a valid evolution equation for Eq. (E.1).

In the general case, we decompose the time interval $[0, 1]$ into $k$ intervals $[t_i, t_{i+1}[$ on which $\theta(t)$ is continuous and the proposed method can be directly extended using an inflation factor $\alpha = k$, introducing $\mathbf{v}_k \in \mathbb{R}^d$. □

Note that the result of this lemma gives an equality between the two flows defined on the *whole* space $\mathbb{R}^d$. The next result is an approximation result which holds on a compact domain $C \subset \mathbb{R}^d$. For a function $f : \mathbb{R}^d \to \mathbb{R}$, we denote $\|f\|_{C,\infty} = \sup_{x \in C} |f(x)|$.

**Proposition 6.** *The* UpDown *model is universal in the class of time-dependent vector fields. Let $C \subset \mathbb{R}^d$ be a compact domain. For every time-dependent vector field (such that it is time continuous and is Lipschitz in space) $w : [0, 1] \times \mathbb{R}^d \mapsto \mathbb{R}^d$ and its associated flow $\varphi(t, \mathbf{x})$ there exist time dependent parameters of the* UpDown *model such that*

$$\begin{cases} \dot{\mathbf{q}}(t) = \theta_1(t)\sigma(\mathbf{v}(t)) + b_1(t) \,, \\ \dot{\mathbf{v}}(t) = \theta_2(t)(\mathbf{q}(t)) + b_2 + \theta_3(t)\sigma(\mathbf{v}(t)) \,, \end{cases}$$

*is $\varepsilon$-close to the solution $\varphi(1, \mathbf{x})$, e.g. $\|\varphi(1, \mathbf{x}) - \mathbf{q}(1, \mathbf{x})\|_{C,\infty} \leq \varepsilon$.*

*Proof.* The proof is standard and we include it here for self-containedness. It is the consequence of [1] and Lemma 5. Let $B(0, r)$ a ball of radius $r$ in $\mathbb{R}^d$ which contains $\varphi(t, \mathbf{x})$ for all time $t \in [0, 1]$. The flow associated with a given time-dependent vector field $v(t, \cdot)$ can be approximated by a vector field which is piecewise constant in time; i.e. let $\varepsilon > 0$ be a positive real, (by continuity in time of $v(t, \cdot)$) there exists a decomposition of $[0, 1]$ into $k$ intervals $[t_i, t_{i+1}]$ and Lipschitz vector fields $v_i(\mathbf{x}) = f(\mathbf{x}, \theta_i)$ where $f$ is the single hidden layer in Eq. (3.7) such that $\|v_i(\mathbf{x}) - v(t, \mathbf{x})\|_{B(0,r),\infty} \leq \varepsilon$ for $t \in [t_i, t_{i+1}]$. Denote by $w(t, \cdot)$ the time-dependent vector field defined by $w(t, \cdot) = v_i(\cdot)$ for all

$t \in [t_i, t_{i+1}]$. Thus, denoting the flow of $v(t, \cdot)$ by $\varphi^v$ and the flow of $w(t, \cdot)$ by $\varphi^w$, we get

$$\|\varphi^v(1, \mathbf{x}) - \varphi^w(1, \mathbf{x})\| \leq \int_0^1 \|v(t, \varphi^v(t, \mathbf{x})) - v(t, \varphi^w(t, \mathbf{x}))\| + \|v(t, \varphi^w(t, \mathbf{x})) - w(t, \varphi^w(t, \mathbf{x}))\| \, \mathrm{d}t$$

$$\|\varphi^v(1, \mathbf{x}) - \varphi^w(1, \mathbf{x})\|_{C,\infty} \leq \int_0^1 \mathrm{Lip}(v)\|\varphi^v(t, \mathbf{x}) - \varphi^w(t, \mathbf{x})\|_{C,\infty} + \|v(t, \cdot) - w(\cdot)\|_{B(0,r),\infty} \, \mathrm{d}t$$

$$\leq \int_0^1 \mathrm{Lip}(v)\|\varphi^v(t, \mathbf{x}) - \varphi^w(t, \mathbf{x})\|_{C,\infty} \, \mathrm{d}t + \varepsilon \,,$$

where $\mathrm{Lip}(v)$ denotes a bound on the Lipschitz constant of $v(t, \mathbf{x})$ w.r.t. $\mathbf{x} \in B(0, r)$ for all $t \in [0, 1]$. Then, the Grönwall lemma [41] gives

$$\|\varphi^v(1, \mathbf{x}) - \varphi^w(1, \mathbf{x})\|_{C,\infty} \leq \varepsilon e^{\mathrm{Lip}(v)} \,. \tag{E.2}$$

By Lemma 5, $\varphi^w(1, \mathbf{x})$ can be approximated by the flow of the UpDown and the result is obtained via the triangle inequality. □

In this section, we focused on a universality result in the space of time-dependent vector fields. Interestingly, due to the additional dimensions, it is likely that the model is universal in the space of functions as well. This conjecture is supported by the quadratic 1D function regression example which shows that the UpDown model is able to capture some maps which are not homeomorphic. We leave this question for future work.

## F  Experimental settings

This section describes our experimental settings. We use our UpDown model for all experiments and simply use a weighted Frobenius norm penalty for all parameters. Specifically, we weigh this penalty for all parameters with $1$ except for, $\theta_3$ which we penalize by $10$. In our experiments, we have observed better convergence properties for higher penalties on $\theta_3$. This might be due to the special role that $\theta_3$ plays in the model as it subsumes a quadratic term in the original derivation of the UpDown model (see §3.2). In all experiments, we also optimize over the affine map from $x(0)$ to $v(0)$ for the data evolution.

**Simple function regression.** We use 500 epochs for all experiments. For all particle-based experiments we freeze the positions of the particles for the first 50 epochs. We use a ReLU activation function and the MSE loss. We weigh the MSE loss by 100 and the parameter norm loss by 1. We use 500 training samples, 1,000 testing samples and 1,000 validation samples and a batch size of 50. Note that for these simple examples there is, in practice, no real difference between the training, testing, and validation data, as the number of samples is large and the domain is $[-1.5, 1.5]$. We initialize the particle positions uniformly at random in $[-1.5, 1.5]$ and draw the momenta from a Gaussian distribution with zero mean and standard deviation $0.1$. All time-integrations are done via a fourth-order Runge-Kutta integrator with time-step $0.1$. For optimization, we use Adam with a learning rate of $0.01$ and the ReduceLROnPlateau learning rate scheduler of PyTorch with a learning rate reduction factor of $0.5$.

**Spiral.** The spiral data is generated between time $t = 0$ and $t = 10$ with 200 uniformly spaced timepoints. Training is only on small time snippets with an approximate length of $0.25$ time-units. Evaluation is on these short time snippets as well as on the entire trajectory by pasting together solutions for these short time snippets, i.e., an individual short solution starts where the previous one ends. Settings for the spiral are the same as for the simple function regression with the following exceptions. We use 1,500 epochs and the step-size for the fourth-order Runge-Kutta integrator is $0.05$. The MSE loss is still weighted by 100, but the parameter norm loss only by $0.01$. We randomly draw 100 new training samples during each epoch and use 100 evaluation samples and 1,000 short range samples and 1 long-range testing sample. All samples are randomly drawn from the trajectory. However, as the trajectory is traversed at highly nonuniform speed the samples are drawn from a uniform distribution across the trace of the spiral. As for the simple function regression experiment, there is little practical difference between the training, validation, and testing data as the problem is so simple. However, this is not of concern in these experiments as the prime objective is to study the fitting behavior of the different models. We use a batch size of 100.

**Rotating MNIST.** We use the data provided by the authors of [40] and follow the same autoencoder architecture, except that our encoder maps into a 20-dimensional representation space. The number of particles is set to 100 and the inflation factor $\alpha$ is set to 10. For optimization, we use `Adam` with a learning rate of 0.001 and the `CosineAnnealingLR` learning rate scheduler of `PyTorch`. We train for 500 epochs with a batch size of 25 and the parameter norm loss set to 0.1.

**Bouncing balls.** As in the rotating MNIST experiment, we rely on the data provided by the authors of [40], follow their autoencoder architecture and set the dimensionality of the representation space of the encoder to 50. The first three images of each sequence are provided to the encoder by concatenating the images along the channel dimension. The inflation factor $\alpha$ is set to 20 and we use 100 particles. We optimize over 100 epoch using `Adam` with the `CosineAnnealingLR` learning rate scheduler of `PyTorch`, the initial learning rate is set to 0.001 and the parameter norm loss is set to 0.0001.

# G   Additional results

**Simple function regression.** In Section 4, we considered approximating a quadratic-like function. Here we show parallel results for approximating a cubic function $y = x^3$. We will also include some additional figures for the quadratic-like regression function. Note that whereas the cubic function is invertible (but not diffeomorphic), the quadratic-like one considered in Section 4 is a simple example of a non-invertible function. Tab. 1 shows the number of parameters for the four different formulations for both regression functions. Fig. 6 shows for the cubic regression the test loss and the network complexity, as measured by the Frobenius norm [28], for the four formulations. On average the particle-based approaches show the best fits with the lowest complexity measures, indicating the simplest network parameterization. Note however that while the dynamic particle approach greatly outperformed the static particle approach for the quadratic-like function (see Fig. 2) this is not the case here. In fact, the static particle approach shows slightly better fits than the dynamic one. This might be because the cubic function is significantly simpler to fit and hence may not benefit as much from the dynamic approach. To illustrate that fitting the quadratic-like function is indeed harder, Figs. 7 and 8 show function fits for different numbers of particles for the cubic function and the quadratic-like function, respectively. All these fits are for the particle-based dynamic `UpDown` model. Clearly, very few particles can achieve reasonable fits for a simple function. As little as two particles already show a good fit for the cubic function, whereas the quadratic-like function requires with more particles. This supports our hypothesis that fitting more complex functions may require more particles.

Since our approach is based on the time-integration of the `UpDown` model it is interesting to see 1) how the mapping is expressed across time and 2) how the parameters, $\theta(t)$, of the `UpDown` model change over time. Fig. 9 shows example mappings for the cubic and the quadratic-like function, respectively. The estimated mappings are highly regular. Lastly, Figs. 10 and 11 show the time-evolutions of the model parameters for the cubic and the quadratic-like function for two different inflation factors. While different parameters show different dynamics, clear changes over time can be observed. In particular, $\theta_2(t)$ and $b_2(t)$ show strong changes. These parameters mostly control the behavior of the hidden high-dimensional state, $v$, as $\theta_3(t)$ is penalized significantly more in our model (see Sec.F) and consequently shows more moderate changes.

**Table 1:** Number of parameters for the simple function regression cubic and quadratic experiments.

| | | Inflation factor | | | | | |
|---|---|---|---|---|---|---|---|
| | **#Particles** | 4 | 8 | 16 | 32 | 64 | 128 |
| static/dynamic w/ particles | 2 | 28 | 52 | 100 | 196 | 388 | 772 |
| | 5 | 58 | 106 | 202 | 394 | 778 | 1,546 |
| | 15 | 158 | 286 | 542 | 1,054 | 2,078 | 4,126 |
| | 25 | 258 | 466 | 882 | 1,714 | 3,378 | 6,706 |
| dynamic direct | n/a | 153 | 461 | 1,557 | 5,669 | 21,573 | 84,101 |
| static direct | n/a | 37 | 105 | 337 | 1,185 | 4,417 | 17,025 |

**Figure 6:** Function fit (15 particles) for cubic $y = x^3$ for 10 random initializations. *Left*: Test loss; *Right*: time-integral of $\log_2$ of the Frobenius norm complexity. Lower is better for both measures. * indicates number of removed outliers (outside the interquartile range (IQR) by $\geq 1.5\times$ IQR); $\alpha$ denotes the inflation factor.

**Figure 7:** Fits for the *cubic* function with inflation factor 16 and for different numbers of particles. Vertical lines indicate particle positions after optimization. While subtle, the figures suggest that using more particles allows for better approximation of the function. This is confirmed by the test loss values in Fig. 6 (bottom left).

**Figure 8:** Fits for the *quadratic-like* function for inflation factor 16 with different numbers of particles. Vertical lines indicate particle positions after optimization. As this function is more complex than the cubic function 2 and 5 particles is not sufficient for a fit. But 15 and 25 particles result in a well-fitting approximation.

**Spiral.** Tab. 2 shows the number of parameters in each of the four formulations for the spiral experiment. This table complements the Table in Fig. 3 which only showed the number of parameters when using 15 particles.

**Figure 9:** Mapping of the *cubic* function (left) and the *quadratic-like* function (right). As can be seen, the mappings are *highly regular*.

**Table 2:** Number of parameters for the spiral experiment.

|  | #Particles | Inflation factor | | | |
|---|---|---|---|---|---|
|  |  | 16 | 32 | 64 | 128 |
| static/dynamic w/ particles | 15 | 1,116 | 2,172 | 4,284 | 8,508 |
|  | 25 | 1,796 | 3,492 | 6,884 | 13,668 |
|  | 50 | 3,496 | 6,792 | 13,384 | 26,568 |
| static direct | n/a | 1,282 | 4,610 | 17,410 | 67,586 |
| dynamic direct | n/a | 6,026 | 22,282 | 85,514 | 334,858 |

**Figure 10:** Weight evolution across time (i.e., continuous depth) for 15 particles when fitting the *cubic* function using the UpDown model: $\dot{\mathbf{x}}(t) = \theta_1(t)\sigma(\mathbf{v}(t)) + b_1(t)$, $\dot{\mathbf{v}} = \theta_2(t)\mathbf{x}(t) + b_2(t) + \theta_3(t)\sigma(\mathbf{v}(t))$. Results are for the dynamic with particles approach. Top: Inflation factor 16. Bottom: Inflation factor 64. Changes in parameter values can clearly be observed.

**Figure 11:** Weight evolution across time (i.e., continuous depth) for 15 particles when fitting the *quadratic-like* function using the `UpDown` model: $\dot{\mathbf{x}}(t) = \theta_1(t)\sigma(\mathbf{v}(t)) + b_1(t)$, $\dot{\mathbf{v}} = \theta_2(t)\mathbf{x}(t) + b_2(t) + \theta_3(t)\sigma(\mathbf{v}(t))$. Results are for the dynamic with particles approach. Top: Inflation factor 16. Bottom: Inflation factor 64. Changes in parameter values can clearly be observed.

## Footnotes

[4]I.e smoothness asks for Lipschitz regularity vector field, which ensures existence and uniqueness of the flow.

[5]The actual Barron norm is defined as the infimum of the r.h.s. in Eq. (B.4) on all the possible representations of the function $f(\cdot, \theta)$ as a single-hidden-layer.

[6]For example, $S$ for our UpDown model simply corresponds to the dimension of its state space: $S = (\alpha+1)d$, where $\alpha$ is the inflation factor and $d$ the data dimension. Note that in our experiments with the UpDown model we also learned an affine map from the initial conditions $\mathbf{x}(0)$ to the initial conditions $\mathbf{v}(0)$. Such a map has $\alpha d(d+1)$ parameters. These parameters are included in the table of Fig. 3 and in Tables 1/2 summarizing the number of mode parameters. However, we will not consider parameters in our discussion here, as they would equally apply to both a shooting and a direct optimization approach and could also be avoided by simply initializing $\mathbf{v}(0)$ to zero. A similar initialization to zero approach is, for example, commonly taken in ResNets when increasing the number of feature channels [20].

[7]In this section, we mean affine with respect to $(\sigma(\mathbf{q}_j(t)))_{j\in 1,\dots,K}$.

[8]Note that the case $\alpha = 1$ is similar in its formulation to a second-order model on $\mathbf{q}$.