[Reviews · NeurIPS 2020]

Review 1

Summary and Contributions: In this paper, the authors describe a shooting formulation of deep neural networks using the optimal control theory language in the spirit of the Large deformation diffeomorphic metric mapping (LDDMM). To this end, they define an Hamiltonian dynamic where the parameters of the method are then a set of initial pairs (position, momentum) called particles (a.k.a "control points" in the LDDMM literature). During the learning step, the particles are updated in order to optimize the generated flow (with respect to a particular task). Finally, new data are flowed with the learnt dynamic to get a response variable.

Strengths: This is nice idea to cast DNN framework into a well grounded mathematical theory. Some theoretical warranties are provided. A substantial piece of python code is also provided. It is well documented and work (almost) out-of-the-box. It seems to be ready to be published online.

Weaknesses: Some key parameters of the methods are discussed rather quickly and not so clearly: The inflation parameter alpha (whose effect remains obscure to me) and how the set of particles are chosen during the experiments (except in the first experiment where it is mentioned that the particles are taken randomly).

Correctness: it seems ok to me

Clarity: Due to the lack of space, the paper is dense to read and the authors go a bit quickly on important points (especially on the numerical experiments). Moreover, the authors introduced too many methods for a 8 pages paper: it will be more suited in a longer journal paper. The python code provided in the supplementary material is quite intricate (I know that the method is rather complex!). But it is hard to quickly get the implementation details not given in the text.

Relation to Prior Work: Nicely written though dense due to the page limit.

Reproducibility: Yes

Additional Feedback: Add an algorithm section or enhance Fig. 1 to provide the reader a big picture of the method. To increase the impact of this contribution as a conference paper, I suggest to focus more on the code and take a more "algorithmic" point of view. Then, I suggest to write a second journal paper with more detailed theoretical contributions. -------------------------------------------- In their answer, the authors promise to refactor the paper to ease the reading. This second version could make this work a nice contribution to the NeurIps conference. That being said, I still would like to know how are initialized the set of particles in the experiments...


Review 2

Summary and Contributions: In this paper the authors propose a novel perspective on parameterizing and training continuous-depth neural networks. The approach taken relies on a couple of ideas and techniques which together make this work quite innovative and different from established deep learning techniques: The authors identify/assert that having constant or piecewise parameterized parameters for continuous-depth neural networks is a shortcoming that is worth solving. Yet, to avoid making the parametrization infinite-dimensional, they propose a “complexity” regularizer on the parameters \Theta(t). While this does not lead to an explicit representation of the parameters, it does constrain their implicit dimensionality if done correctly. By considering the adjoint sensitivity method for the regularized continuous depth dynamics, the authors recognize and describe a set of optimality constraints that need to be fulfilled for \Theta(t) to minimize the training loss. Finally, instead of solving for solutions of these optimality constraints directly, they parameterize the solutions with a “shooting approach”: The solutions for \Theta(t) are given by boundary conditions (at t=0) of a set of particles and their first temporal derivative. The adjoint method places further constraints on these particles, including the boundary conditions at time T such that these implicitly parameterize a solution for \Theta(t). The number of particles used effectively control the number of parameters and the allowed complexity of the function described by the model.

Strengths: The paper describes an innovative approach to parameterize and train a continuous-depth neural ODE while at the same time regularizing the solution to limit its ‘complexity’. The approach and the derivation seem correct to me. Throughout the paper the reader is exposed to the shooting method, a generally well known method in numerical analysis but which has not found any previous applications in deep learning; and to a novel perspective how it can be applied to neural ODEs. The experimental section contains clear evidence that the approach works in general, and specifically that the indirect parametrization of the time-variant Theta(t) via a particle based shooting approach leads to results that can outperform more traditional approaches when given otherwise similar model-architectures and constraints. I also think that the experimental section contains reasonable experiments, including ablation studies and (small) analyses which is at this point probably more interesting than chasing state of the art results.

Weaknesses: While the experimental section does contain evidence that the method works and produces reasonable results on comparably simple problems; it is not obvious that this approach comes with practical benefits and leads to improved results compared to more traditional parametrizations and methods which are often more flexible and appear considerably simpler.

Correctness: The arguments, derivations and general construction of the method seems correct. I did not check all the derivations thoroughly though; but only went over some of the derivation in detail.

Clarity: The paper is well written and does a good job introducing the various innovations presented in this work. The appendix contains a wealth of additional information, both theoretical and crucial practically relevant comments and derivations. Because the innovation presented in this work is mainly conceptual, I think it was a good choice to leave some of these aspects to appendix. Even though some of them are practically very relevant.

Relation to Prior Work: As far as i can tell the paper discusses all relevant prior work adequately.

Reproducibility: Yes

Additional Feedback: *** Author feedback update *** I would like to thank the authors for the clarifications in their rebuttal. I will maintain my recommendation to accept the paper. Neither reading the other reviewers comments, nor reading the rebuttal changed my overall assessment of the paper. Thanks everyone, ----- Line 5: It’s not clear to me why having constant parameters throughout makes a model not “continuous-depth”; I don’t understand the statement in line 106 with 3d images vs R^d (with d the dimensionality of the input like in line 109?) Line 120: if R(\Theta) is some corresponding norm?


Review 3

Summary and Contributions: The work "A shooting formulation of deep learning" mostly extends on the work of "Neural ordinary differential equations" [10]: Similarly, this work reasons about a continuous version in time of ResNet-like deep neural network architectures through ordinary differential equations (ODEs). I've identified three major contributions on top of [10]: - The work re-parametrizes the fitted gradient function f of the ODE through a shooting formulation: the function is solely _implicitly_ characterized by initial state and momentum pairs and the applied regularization of the parameters over time. How these are interpreted depends on the underlying functional model of f. So training does not fit weights of some estimator, but the initial moment (and later the initial state for the restricted ensemble) of the Hamiltonian samples. - In contrast to previous work, the parameters theta that govern the function f (the gradient of the ODE formulation) can become time dependent. As the functional form of f is implicitly defined, there is no time discretization of the parameter function theta(t). - To solve efficiency issues with the shooting formulation (that is, initially every training sample would be attributed with an additional momentum weight that is fitted during training), the work proposes to optimize a limited ensemble of state-momentum terms. The larger this set of representer points the more parameters can be tuned during training.

Strengths: - Continuing the discussion about continuous deep neural networks is a relevant problem. - Allowing for continuously changing parameters theta(t) to the ODE function, instead of static theta without any time discretization. - Deriving an efficient ODE formulation that mimics ResNet layers (the work calls this the UpDown model): introduction of a limited ensemble of Hamiltonian samples and the introduction of the additional state v to end-up with a model that is linear in theta. - The theoretical results enrich the understanding and discussion about continuous time deep estimators. Even though, the results of this work might not be directly applicable to any practical machine learning task today (e.g., a convolutional layer formulation is not included in this work), I think it might inspire future research. Personally, I took quite some interesting points from this work. The shooting formulation opens a different viewing angle on the parametrization of neural ODEs. - The empirical results show an improvement over the static theta parameter models for a set of toy problems. - Comes with a public implementation.

Weaknesses: - The size of each of the experiments feels toy-ish. - It might not be straight forward to apply the shooting formulation to different neural network layers. One of the big advantages of [10] is, that it connects quite naturally to established deep discrete estimators to parameterize the dynamics of the ODE. This is not true here. The authors do point out that the autodiff machinery helps to explore different layer operators (appendix C), however this will not be as straight forward as combining layers without making the nature of the layers explicit [10]. In some sense, I think there is a trade-off to be had: either there is no real knowledge of what kind of ResNet update is learned in [10] but with a fixed theta over time, or an explicit algebraic form as in eq. 3.7 with changing theta over time. I do understand that this work adds the additional benefit of having dynamic parameter functions, but I wonder if the shown boost in performance in the experiments transfers to bigger problems. - The supplementary material included a snapshot of the code that was used by the authors to produce the results in the paper. I do believe that it does not include all the experiments though. Maybe it makes sense to include those as well? However, there are enough details in the paper to reproduce the experiments with the implementation being publicly available.

Correctness: Yes, I believe the claims and methodology in this work are correct. However, I think there might be something wrong with the second part of equation 3.9. I do understand the discussion in the appendix (line 620) that derives the final form of the up-down model. However, it seems hard to match that with 3.9. E.g., how come theta_2 reappears in 3.9 without its derivative? For the sake of understanding, I think it would make sense to have a more detailed discussion about the introduction of additional parameter functions like theta_3. This is not really mentioned in the monograph. (And for completeness, it makes sense to add the function parameter t, i.e., v(t), to the second part of 3.9 as well.)

Clarity: - The paper is mostly well written. There are some aspects, where the work feels dense. However, there are multiple pointers to the supplementary material that come with additional examples, interpretations and proofs. It is worthwhile to have that handy while reading the main monograph. - Equations in the appendix seem not to be numbered correctly. - The definition and use of the inflation factor (line 190) maybe deserves one more comment. It took me a moment to notice the alpha again in line 191. Footnote 4 in the appendix helped to understand it even better.

Relation to Prior Work: Yes, this work is careful about stating the relation to prior work.

Reproducibility: Yes

Additional Feedback: - How is this work connected to kernel machines? In particular, as A.2 mentions the connection to reproducing kernel Hilbert spaces. - Can a time-dependent theta also be simulated by increasing the state space for static direct, which is closest to the original work of neural ODEs [10]? - How much more expensive in run-time is the proposed method? Although, only a forward ODE solver run is required, the method needs to track each particle along the way. ===================== I would like to thank the authors for their feedback. I am excited about this idea and I am hopeful that further research will answer and tackle the additional questions that this work opens up.

[Author Response · NeurIPS 2020]

*We thank all the reviewers for their positive feedback and constructive comments. Below, we first discuss two issues*
*raised by multiple reviewers and then answer and clarify reviewer-specific comments / questions.*

**Inflation factor $\alpha$**: **R1** and **R4** asked for clarification of the inflation factor $\alpha$. It is simply the increase in feature
dimensions for the UpDown model. I.e., if the dimension of $q_1$ is $d$ then the dimension of $q_2$ is $\alpha d$. We will clarify this.

**Practical implications**: **R3** and **R4** were curious whether our experiments demonstrate practical benefits of our
approach. We show state-of-the-art results for rotated MNIST (Fig. 4). However, we have not tackled large-scale
learning tasks yet. Our primary contribution is, as acknowledged by **R4**, to *"enrich the understanding and discussion*
*about continuous time deep estimators"* and to offer *"a different viewing angle on the parametrization of neural ODEs."*
We are in the process of extending our approach to convolutional networks and to ResNet-like structures with striding
and feature-dimension changes. This will allow us to explore the performance of our approach for larger-scale tasks.

*Response to **R1***

**Paper too dense**: This is a fair point. Our aim was to make the main manuscript as self-contained as possible. Hence,
we moved many details to the supplementary material. We will, as suggested, add to Fig. 1 an algorithmic description.

**Code complexity**: We will add a more detailed README file and will supplement the existing code with Jupyter
notebooks so that a reader can more easily follow. We will also add a simple PyTorch example illustrating the approach
in parallel with the new algorithmic description and independent of the more complex implementation we submitted.

*Response to **R3***

**"Line 5: It's not clear to me why having constant parameters throughout makes a model not continuous-depth"**
We will rephrase our sentence. The static parameters setting is indeed continuous-depth but obviously does not leverage
time varying parameters. **"I don't understand the statement in line 106 with 3d images vs $\mathbb{R}^d$."** In the LDDMM
community, the deformation map operates on 2D or 3D spaces, whereas in the DL setting, the map is usually defined in
much higher dimensions, e.g. the number of pixels of an image or for us the state-space of our UpDown model. **Line**
**120: Is $R(\Theta(t))$ some corresponding norm?** In our current setting, $R(\theta(t))$ is the Frobenius norm. However, $R(\theta(t))$
can be more general (see, e.g., Appendix A.2 on the Barron norm). We will clarify these points in the manuscript.

*Response to **R4***

**"... a convolutional layer formulation is not included"** We had, in fact, already implemented shooting for convo-
lutional layers (it is part of the submitted code), but did not include this material and associated experiments to
avoid making the paper even more dense. **"It might not be straight forward to apply the shooting formulation to**
**different neural network layers ..."** We agree that there is a trade-off between the ease with which [10] can be
deployed for $\theta$ constant in time and the increased complexity of our approach for handling dynamic $\theta$. In the "Rotated
MNIST" experiment, e.g., our formulation clearly shows benefits over its non-dynamic (stat. direct in Fig. 4)
counterpart. Hence, increased complexity can improve performance and can therefore be beneficial. Note also that our
implementation allows us to automatically derive the shooting equations (Appendix C and implementation) if desired.

**"How is this work connected to kernel machines? In particular, as A.2 mentions the connection to reproducing**
**kernel Hilbert spaces."** Since our space of vector fields is finite dimensional, it is a proper RKHS when endowed with
the Frobenius norm. This connection still needs further exploration using other perspectives present in the DL literature.
**"Can a time-dependent $\theta$ also be simulated by increasing the state space for static direct, which is closest to the**
**original work of neural ODEs [10]?"** This is an excellent question. Unfortunately, we do not have a theoretically
conclusive answer at this point. **"How much more expensive in run-time is the proposed method? Although,**
**only a forward ODE solver run is required, the method needs to track each particle along the way."** Currently,
depending on the number of particles, the method is more expensive in terms of run-time compared to [10]. However,
our primary focus was on functionality at the current stage, leaving substantial room for optimization and making
runtime comparisons difficult at this point. E.g., the matrices in the UpDown model are obtained via outer-products.
Hence, explicitly forming these matrices could be avoided. Further, for convolutional formulations the particles do not
need to have the same dimension as the input images, but could be much smaller patches. For the experiments in the
manuscript the static direct approach (which does not use any particles) indeed runs faster, but not dramatically so.

**"... [the supplementary material] does not include all the experiments ..."** We decided to omit any code that would
require reviewers to download additional data (e.g., for the "bouncing balls" experiment). Code for *all* experiments
will be included in the final version. **"... something wrong with the second part of equation 3.9."** It is actually a
renaming of the variables – we overloaded the notations. We will fix this. **"Equations in the appendix seem not to**
**be numbered correctly"** We apologize that equation numbers in the appendix can easily be confused with section
numbers. We will fix this in the final version. **"... more discussion [of broader impacts] is needed."** We will add
more discussion related to the societal impacts of the algorithms that may result from our work.

[Meta-Review · NeurIPS 2020]

The paper presents a new perspective on parameterizing and training continuous-depth neural networks from the vantage point of control theory. The paper can be understood as an extension of the neural-ODE line of work. In particular they consider a particle based parameterization of a neural-ODE that can be solved for via a shooting method. All reviewers agree that this is an innovative, new, approach that is theoretically sound and that the paper is well written with good coverage of the shooting method (for the unfamiliar reader). The provided implementation further aids reproducibility. The only small weakness is that the experiments in the paper are largely on small-scale problems, and it is thus not completely obvious whether the presented approach will yield improvements over NODE in other scenarios. Overall this is a good paper that is well motivated, well written and makes an interesting theoretical connection. It should clearly be accepted.